# Black carbon and dust alter the response of mountain snow cover under climate change

Marion Réveillet [1] ✉, Marie Dumont [1] ✉, Simon Gascoin [2], Matthieu Lafaysse[1], Pierre Nabat[3], Aurélien Ribes [3], Rafife Nheili[1], Francois Tuzet[1], Martin Ménégoz[4], Samuel Morin [3], Ghislain Picard[4] & Paul Ginoux[5]

By darkening the snow surface, mineral dust and black carbon (BC) deposition enhances snowmelt and triggers numerous feedbacks. Assessments of their long-term impact at the regional scale are still largely missing despite the environmental and socio-economic implications of snow cover changes. Here we show, using numerical simulations, that dust and BC deposition advanced snowmelt by 17 ± 6 days on average in the French Alps and the Pyrenees over the 1979–2018 period. BC and dust also advanced by 10-15 days the peak melt water runoff, a substantial effect on the timing of water resources availability. We also demonstrate that the decrease in BC deposition since the 1980s moderates the impact of current warming on snow cover decline. Hence, accounting for changes in light-absorbing particles deposition is required to improve the accuracy of snow cover reanalyses and climate projections, that are crucial for better understanding the past and future evolution of mountain social-ecological systems.

Snow cover changes have drastic consequences for numerous components of the Earth system such as water resources[1,2], ecosystems[3,4] and the Earth climate through complex feedbacks[5]. In response to climate change, the snow cover duration (SCD) in mountain areas has declined at the global scale since the 1950s, on average by 5 days per decade at low elevations[6,7]. As a response to snow cover decline and enhanced snowmelt, the peak runoff from snowmelt has shifted by around 10 days on average over the 1965–2005 period for the Alps[8]. Changes both in the magnitude and timing of the snowmelt runoff threaten the availability of water resources in downstream regions, and in particular for irrigation[2]. Snow cover trends are mostly attributed to changes in atmospheric drivers, and especially to the warming that intensified over the last decades to reach 0.3 ± 0.2 °C per decade for mountainous areas at global scale[6].

Light-absorbing particles (LAPs) such as black carbon (BC), brown or organic carbon, mineral dust and algae are potent drivers of snow

cover changes[9–11]. LAPs darken the snow surface when deposited on the snow cover and amplify several snow-albedo feedback loops, drastically modifying the snow cover evolution and duration[12]. LAPs indeed accelerate the coarsening of the snow microstructure, leading to more solar energy absorption and to an acceleration of the intrinsic snow albedo feedback[13]. Moreover, LAPs are partially retained at the snow surface during melt, amplifying the decrease in the albedo[14]. The impact of LAPs on the snow cover is, thus, modulated by meteorological conditions, leading to complex interplays between the LAPs and the meteorological drivers of the snow cover evolution[5].

BC is the most efficient absorbing aerosol in the atmosphere[12,15] and its deposition strongly impacts snowmelt rates[12], as shown for instance in High Mountain Asia, Europe, North and South America (ref. 16 and references therein). Yet the contribution of the regional changes in BC deposition to the observed snow cover decline remains to be assessed[6]. In central Europe, an increase in BC deposition since 1850

[1]Univ. Grenoble Alpes, Université de Toulouse, Météo-France, CNRS, CNRM, Centre d'Etudes de la Neige, 38000 Grenoble, France. [2]Centre d'Etudes Spatiales de la Biosphère (CESBIO), Université de Toulouse, CNRS/CNES/IRD/INRAE/UPS, 31400 Toulouse, France. [3]CNRM, Université de Toulouse, Météo-France, CNRS, Toulouse, France. [4]Univ. Grenoble Alpes, CNRS, IRD, IGE, 38000 Grenoble, France. [5]NOAA Geophysical Fluid Dynamics Laboratory, 201 Forrestal Road, Princeton, NJ 08540, USA. ✉e-mail: marion.reveillet@univ-grenoble-alpes.fr; marie.dumont@meteo.fr

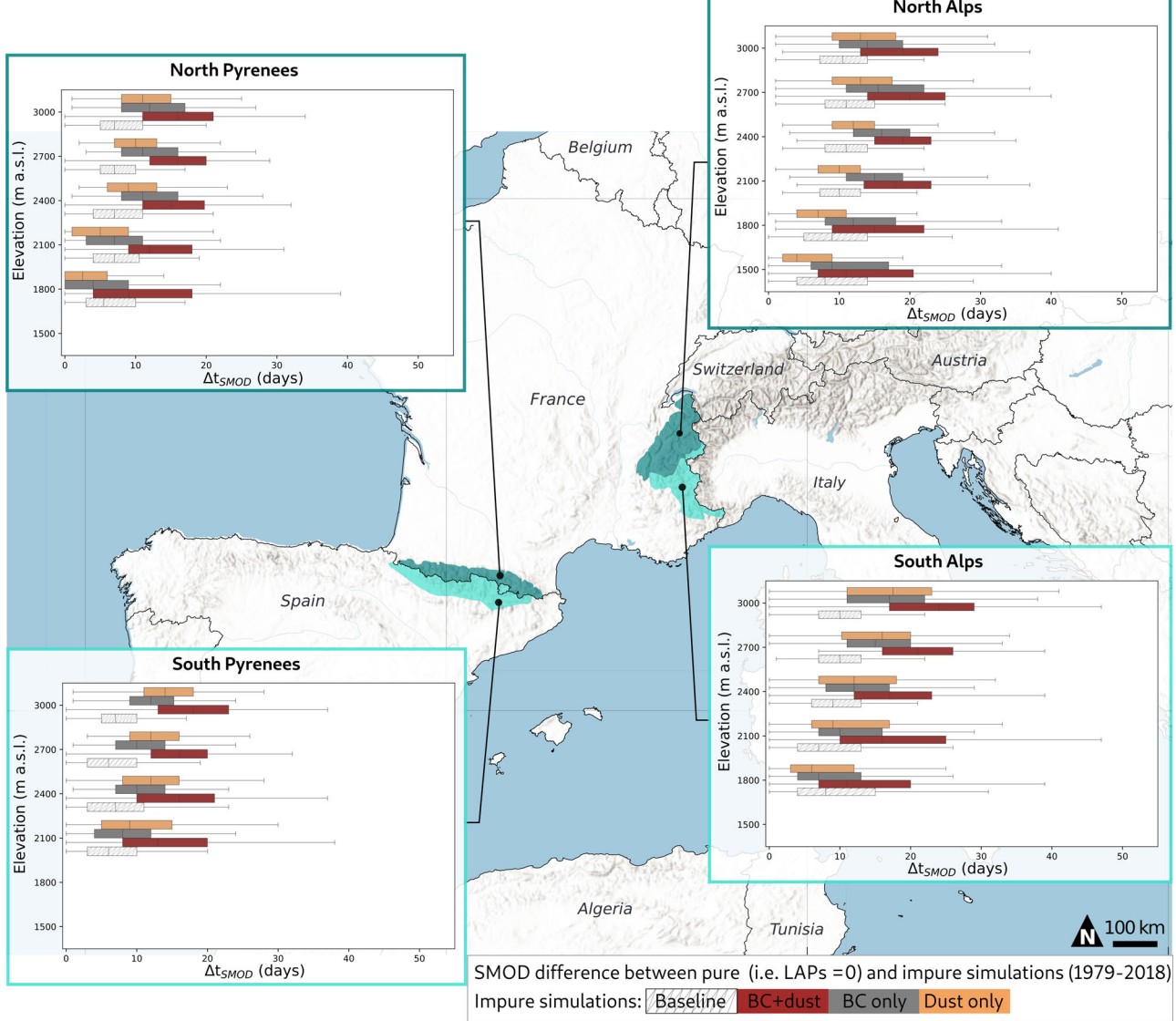

**Fig. 1 | Shortening of the snow season related to BC and dust deposition.** This shortening is computed as the difference in snow melt-out date (SMOD) between the pure snow simulations and all the configurations: simulations considering BC and dust (red), only BC (gray), only dust (orange), considering implicitly the LAPs (i.e., an albedo decrease based on the snow age only (white, hatched)). SMOD differences are computed as a function of the elevation, computed considering the entire study period (i.e., 1978–2018), for the North and South Pyrenees (left) and the North and South Alps (right). The boxes show the quartiles of the distribution corresponding to the inter-annual and spatial variability. Minimum/maximum ranges (excluding outliers) are indicated by the whiskers. Only elevations with a mean SD > 30 cm over the winter period (i.e., 1st of December to 30 of April) are represented.

has been observed until the 1970s due to anthropogenic activities[17]. It has been suggested that BC deposition contributed to the triggering of glacier retreat in the Alps[17]. However, BC deposition has decreased since the 1980s[16,18], due to a reduction of emissions likely primarily to traffic emission decrease[19]. However, the impact of this recent negative trend on snow cover has not been quantified hitherto.

Even if mineral dust deposition events are generally more episodic than BC deposition[20,21], they can have a widespread visible impact on the snow surface (e.g., in Feb. 2021[22]), with a prevailing radiative effect compared to BC[20,23]. In Europe, mountain ranges are generally affected by dust deposition events originating from the Sahara, which contribute to 50–70% of the total annual dust deposition[24]. Yet, the strong spatial and temporal variability combined with the lack of long-term data make it difficult to identify trends over the last decades[25].

Regional assessments of the combined effect of LAPs deposition on snow cover changes over long time periods are currently missing[6,12] and require an accurate representation of the snow-

albedo feedback triggered by the presence of LAPs. The aim of this study is thus to quantify the combined effect of BC and dust deposition on snow cover dynamic and trends at the regional scale of the French Alps and Pyrenees mountain ranges, for the 1979–2018 period. The snow cover represents a major driver of hydrological processes in both mountain ranges. Its variability exerts a strong influence on water resource availability in southern France and northern Spain but also on the income generated by winter tourism in both countries. Considering the large uncertainty in brown carbon deposition and absorption efficiency[12], the effect of brown carbon is excluded from the analysis as well as the effect of algae that generally bloom only at high elevation and late in spring[26], (see the "Limitations" section). Our analysis is based on numerical simulations performed with the detailed snow cover model Crocus accounting for the complex interactions between LAPs and snow[27]. Hence this study exemplifies the impact of dust and BC on seasonal snow cover in mid-latitudes mountain regions.

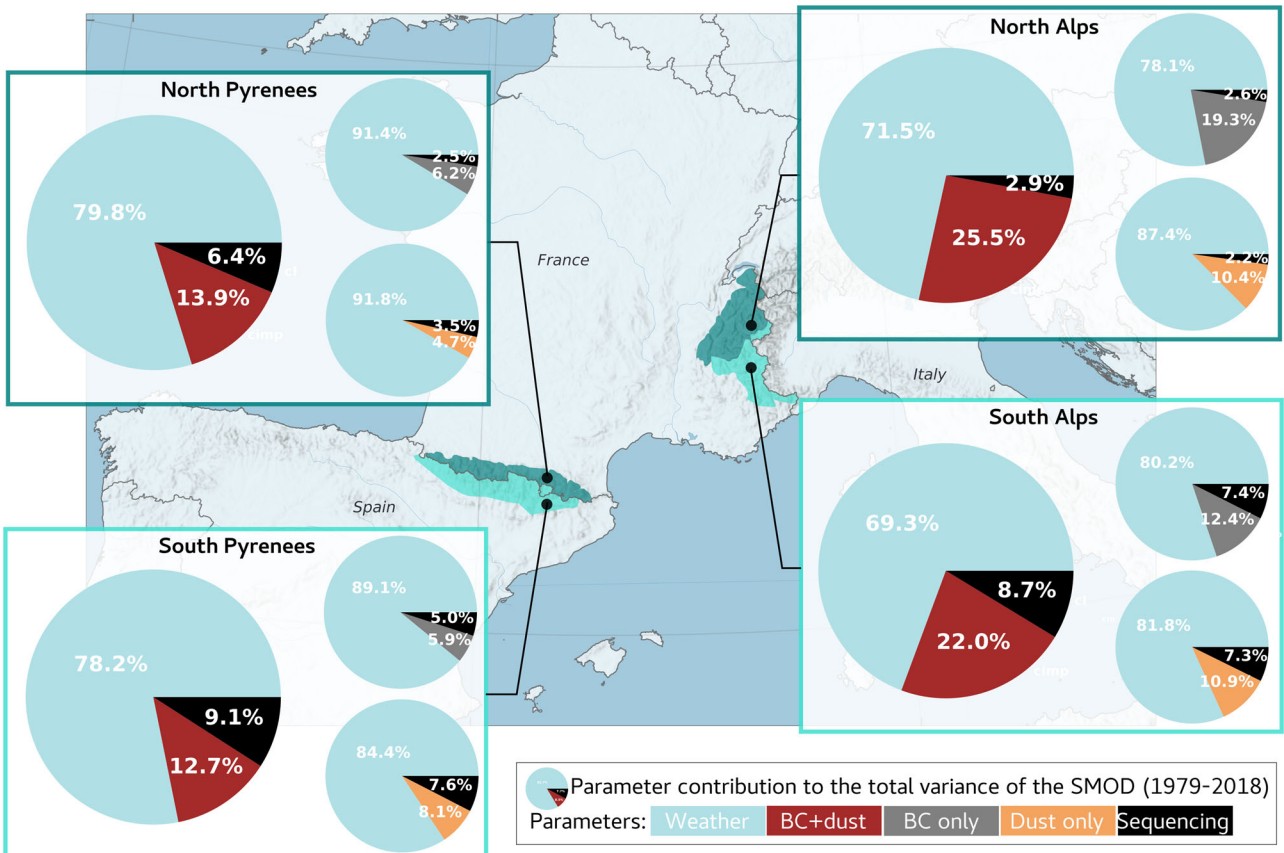

**Fig. 2 | Contribution of the meteorological conditions and BC and dust to the variance of the SMOD.** Contributions computed over the 1979–2018 period at 2100 m a.s.l.. for the North and South Pyrenees (left), North and South Alps (right). Larger circles indicate contributions of the parameters considering the role of BC + Dust (red) and smaller circles represent the contribution of BC (gray) and dust (orange) separately. The sequencing term (black) indicates the contribution of BC and dust and meteorological forcing, and is due to the dependence of the BC and dust contribution to the meteorological conditions.

## Results

Simulations explicitly accounting for dust and BC deposition fluxes are compared to a pure snow simulation, excluding any LAP. Simulations are driven by the hourly meteorological reanalysis S2M[28] and hourly deposition fluxes of BC and dust from the regional climate model CNRM-ALADIN63 driven by reanalysis data, so as to follow the unfolding of observed meteorological conditions[29]. To quantify the impact of BC and dust on snow cover evolution we used the snow melt-out date (SMOD), defined as the last date of the annual longest period with at least 30 cm of snow. This indicator is relevant for water resources and ecosystem studies[6]. The snow cover simulations were evaluated against satellite observations of the snow cover area (SCA) and in situ observations of snow depth at 495 stations (see "Methods" section). Simulations accounting for BC and dust systematically lead to better scores compared to pure snow simulations, with an overall SMOD bias of −0.68 days, an RMSE of 19.2 days and an MAE of 13.3 days at the measurement stations (see "Methods" section).

### Shortening of the duration of the seasonal snow cover due to BC and dust

The comparison of the pure snow simulations with the simulations accounting for BC and dust shows that, by darkening the snow surface, the combined effect of BC and dust leads to an earlier SMOD by 17.8 and 16.1 days on average in the French Alps and Pyrenees, respectively, over the 1979–2018 period (Fig. 1). These reductions correspond to 9.1 and 8.8% of the mean annual SCD. These averages are calculated accounting for the relative surface at each elevation band in both mountain ranges. At local noon, BC and dust decrease the mean surface albedo by 0.03–0.05 on average depending on the area

(Supplementary Fig. S1) and increase the solar energy absorbed by the snowpack by 7 to 13 W m$^{-2}$ on average (Supplementary Fig. S2).

The SMOD reduction due to BC and dust deposition (i.e., ΔSMOD on Fig. 1) varies from 0 to 47 days. The magnitude of the SMOD reduction shows strong inter-annual variations (Supplementary Fig. S3), due to the strong inter-annual variation of BC and dust deposition rates (Supplementary Fig. S4). The SMOD reduction also depends on the location and the elevation (Supplementary Figs. S5 and S6). A larger absolute impact of BC and dust on the SMOD in number of days is found at higher elevations. This is related to later melt and longer snow season, where dust and BC are exposed to higher incoming solar radiation, amplifying its effect on snowmelt acceleration. This mechanism was deciphered using snowpack simulations by ref. 30.

The effect of BC is likely prevailing in the northern part of the French Alps, where higher BC deposition is observed (Supplementary Fig. S4). The differences in ΔSMOD between the BC only and dust only simulations are similar to the uncertainties caused by the uncertainties on dust optical properties of dust (see "Methods" section). The simulated SMOD when accounting for BC is earlier on average by 14.2 days (7.1%) in the North Alps and by 13.3 days (6.9%) in the South Alps, compared to the pure snow simulation. The southern parts of both mountain ranges are more exposed to southerly winds carrying Saharan dust (Supplementary Fig. S4), hence exhibit larger dust depositions than their northern counterparts.

Figure 1 also shows the SMOD reduction between the pure snow simulation and a simulation named "baseline" that accounts only implicitly for the LAPs by decreasing albedo as a function of time since the last snowfall without any link to the actual LAP concentration. This simple treatment of the radiative impact of LAPs, only represented by a

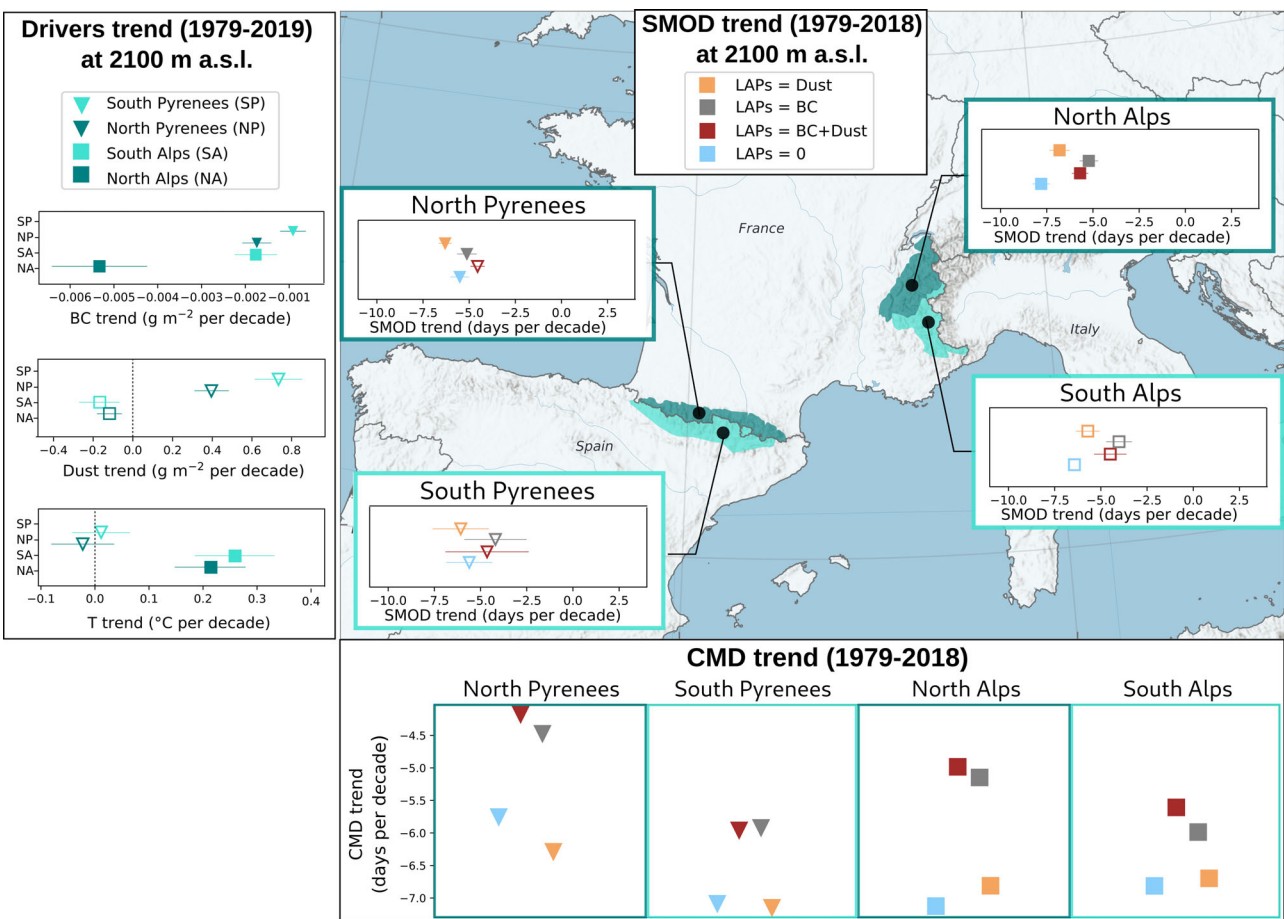

**Fig. 3 | Trends in air temperature (at 2 m above the surface), BC and dust deposition, SMOD and center of mass date (CMD) over 1979–2018.** Trends are from temporal series computed from annual values at 2100 m a.s.l.. Trends are represented as the best estimate and 90% confidence range, per area (North Alps, South Alps, North Pyrenees and South Pyrenees). Only markers of significant trends (*t*-test 0.05) are filled in. Error bars represent the spatial variability. Only elevations with a mean SD > 30 cm over the winter period (i.e., 1st of December to 30 of April) are represented.

darkening of the surface snow over time, is the most common practice in snow modeling[31] but has the disadvantage of neglecting spatial or temporal variations of the LAPs deposition fluxes (see "Methods" section). The simulated SMOD in the baseline configuration is later compared to the explicit simulation on averaged by 7.3 days for the Alps and 9.3 days for the Pyrenees (Fig. 1). This suggests that the temporal and spatial variability of the BC and dust deposition fluxes likely has an important effect on the SCD.

### BC and dust effect on snow cover inter-annual variability

Based on an analysis of the variance (ANOVA[32], see "Methods" section and Supplementary text), we quantify the contribution of the different atmospheric drivers to the SMOD inter-annual variability (see "Methods" section). The combined effect of the BC and dust explains up to 30.7% of the variance of the SMOD at 2100 m a.s.l. in the Alps (Fig. 2, total variance of 329 d²) and up to 21.8% in the Pyrenees (for a total variance of 578 d²). We discuss the results at 2100 m a.s.l, since 20% of the surface of the French Alps and 40% for the Pyrenees lies in this elevation band, making it particularly relevant for water resources and ecosystem disturbances.

For given deposition fluxes of BC and dust, the impact on the snow cover depends on the meteorological conditions[33]. For instance, the same amount of BC and dust deposited at the snow surface has a stronger impact if a long time period without snowfall follows the deposition, than when a new snowfall buries the particles immediately after deposition. Using the ANOVA method, we can separate two types of contribution to the SMOD variance, the

contribution of LAPs deposition observed for average weather conditions, and the contributions through interactions between meteorological conditions and LAP deposition. The later is named the sequencing part (see Supplementary text). This sequencing of meteorological conditions and BC and dust deposition contributes to 2.9–9.1% to the total variance, while the deposition only contributes to 12.7–25.5% of the total variance (Fig. 2). Hence, deposition of dust and BC markedly contribute to the variance of the SMOD and this is partly due to the sequencing between the meteorological conditions and the deposition.

The BC and dust contribution to the SMOD inter-annual variability is larger in the Alps than in the Pyrenees. This is due to the combined effects of three different factors: lower BC deposition (Supplementary Fig. S4), earlier snowmelt and slightly higher inter-annual variability of the meteorological conditions in the Pyrenees than in the Alps (Supplementary Fig. S3). BC deposition contributes to 19.3% and 12.4% to the total variance of the SMOD for the North and South Alps, and to 6.2% and 5.9% for the North and South Pyrenees, respectively. BC contribution to the SMOD inter-annual variability is generally higher than dust contribution, except for the South Pyrenees, more affected by episodic dust outbreaks[34].

In summary, our results show that BC and dust are significant drivers of the SMOD inter-annual variability. The calculation of their contribution has been performed on the 39 years time series without detrending thus includes the effects of trends over the last four decades in the input atmospheric variables, such as air temperature, BC and dust deposition fluxes.

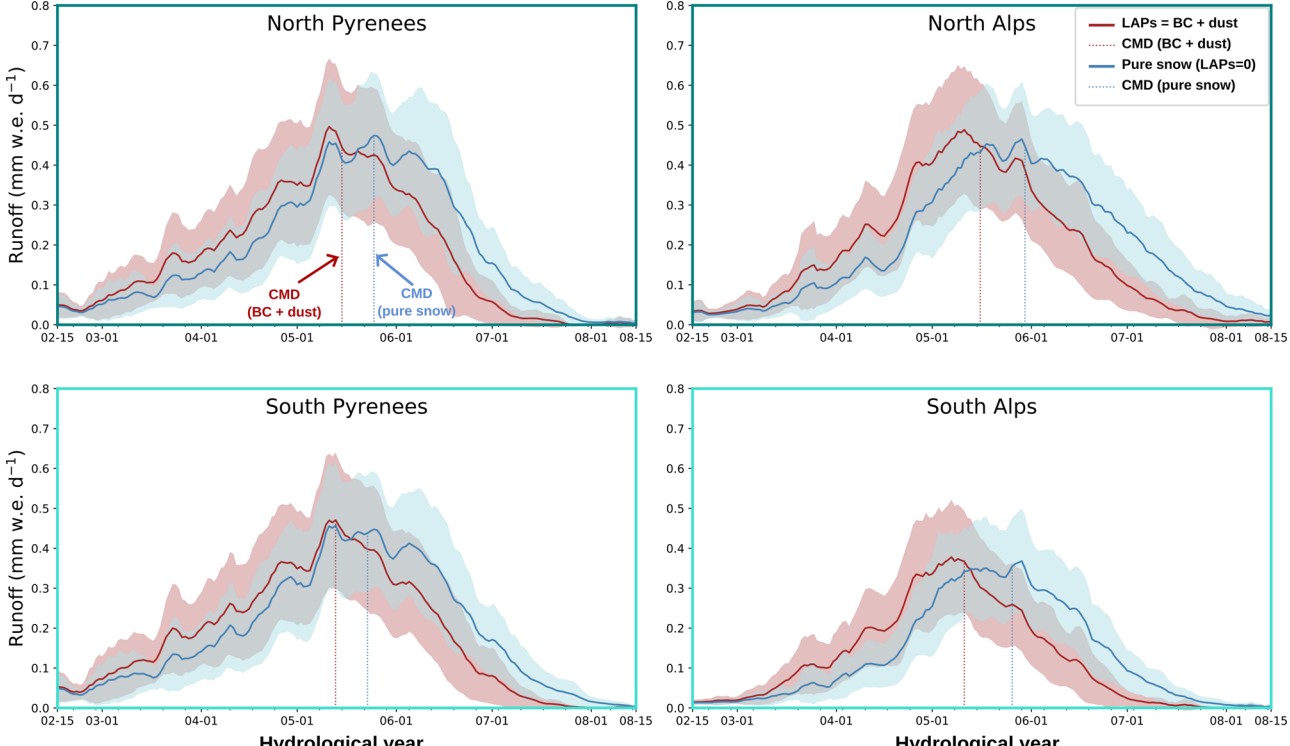

**Fig. 4 | Runoff from snowmelt averaged over the period 1979–2018 (solid line) with the standard deviation (shaded areas) representing the annual variability.** Pure snow simulations are represented in blue and simulations considering the effect of BC and dust are in red. The center of mass date of each simulation, defined as the date for which half of the total annual runoff from snowmelt is reached, are indicated by the dashed lines.

## Past trends (1979–2018)

The simulated SMOD for snow with BC and dust shows a negative trend over the last decades (1979–2018) indicating earlier snowmelt (Fig. 3). In this section, we mostly discuss the results in the North Alps since this is the only area where most of the trends are significant. In the northern Alps, the SMOD at 2100 m a.s.l. decreases by 5.7 days per decade (corresponding to a decrease of 2.9% of the snow season duration), which is consistent with ref. 7. This decrease is consistent with the atmospheric warming rate. This warming rate ranges between +0.25 and +0.3 °C per decade according to the reanalysis data used in this study (Fig. 3).

The simulated SMOD trend for pure snow is more pronounced than for snow with BC and dust. At 2100 m a.s.l., the pure snow SMOD decrease is 7.8 days per decade for the North Alps. The deposition of dust and BC thus alters the response of snow to warming. First, the sensitivity of SMOD to meteorological conditions is not the same for pure snow and for snow containing LAPs (e.g., Fig. 8 in ref. 30). Second, the dust and BC inputs were not constant over the last four decades. BC deposition exhibits a negative trend, with a decrease of 11% on average considering all the massifs. Simulations accounting for BC deposition yield a negative SMOD trend lower than the SMOD trend of the pure snow simulation (Fig. 3). This means that, at the beginning of the study period (1979–1988), the shortening attributed to BC deposition alone (i.e., 15 days on average for all the areas) is higher than for the recent period (2009–2018) (i.e., 10 days only in average). In this recent period, the deposition of BC is reduced compared to the 1980s. BC deposition accelerates melt less than in the 1980s. Thus, the reduction of BC deposition fluxes partly offsets the effect of rising temperature in the 2009–2018 period. Conversely, no significant trend in dust deposition is identified. Dust deposition consequently affects the SMOD trend less than BC deposition does (Fig. 3).

Hence, the deposition of BC and dust modifies the SMOD trend in response to climate warming. In the North Alps, the primary driver of the trend modification is the decrease in BC deposition since the 1980s. This decrease compensates part of the impact of warming on the trend of the SCD at 2100 m a.s.l. and is also observed for all the Alps and Pyrenees to a lesser extent (Supplementary text and Fig. S7). However, the trends presented in this section originate from a single model and the limitations of this study are discussed below.

## Impacts on mountain hydrology

By exerting a strong control on the snow cover ablation rate, BC and dust deposition also has a profound influence on melt water runoff. To quantify this effect, we use the Center of Mass Date (CMD), defined as the date for which half of the total annual runoff from snowmelt is reached[35]. BC and dust lead to a shift in the simulated CMD by up to 15 days earlier in the season (Fig. 4) over the 1979–2018 period compared to pure snow simulations. The effect is larger in the Alps (CMD shift of 15 days) than for the Pyrenees (CMD shift of 10 days) in agreement with the larger effect on SCD in the Alps mentioned above. No significant change in the intensity of the peak runoff can be attributed to the presence of BC and dust (Fig. 4). By increasing the absorbed solar radiation, LAPs cause snowmelt to occur earlier in the snow season, when the available energy is lower (lower shortwave and longwave radiation, lower sensible heat flux). We hypothesize that this is the cause for the unchanged peak runoff magnitude despite the presence of LAPs on the snow surface. A similar mechanism causes a reduction of snowmelt rates under atmospheric warming[36]. Even if the magnitude of the peak runoff is preserved, earlier snowmelt can have a profound consequences on the management of the water resource and downstream alpine ecosystems[37]. The combined effect of BC and dust explains up to 5.4 days of the CMD variance (37% of the total variance,

Supplementary Fig. S8). This is in agreement with the contribution of BC and dust found for the SMOD variance with a prevailing contribution of BC deposition on the northern ranges and dust deposition in the South ranges. The observed changes in the timing of the peak runoff attributed to climate change over the 1965–2005 period are of around 10 days for the Alps[8]. The simulated CMD shift of 10 to 15 days attributed to the presence of BC and dust here is thus comparable in magnitude.

Finally, our results indicate that the shift of the CMD due to warming would be stronger if not compensated by a decrease in BC deposition over the 1979–2018 period (Fig. 3). When considering only BC deposition, the trend in CMD would decrease by 1.3 days per decade on average, with a larger impact (of 2 days per decade) in the North Alps, explained by a stronger negative trend in BC deposition (Fig. 3).

## Discussion

### Relevance and implications

This study analyses the impact of dust and BC deposition on seasonal snow cover variability and trends in the French Alps and Pyrenees, using regional meteorological and deposition reanalysis and the detailed snow cover model Crocus. We demonstrate that BC and dust have a major role in advancing the snow cover melt, with a strong influence on the inter-annual variability of the snow cover and drastic implications for water availability timing. We also show that the response of the snow cover evolution to climate change is strongly modulated by the BC and dust deposition fluxes, suggesting that trends in snow cover and LAPs deposition cannot be investigated independently.

Our results also show that the impact of BC and dust on SMOD depends strongly on elevation, in terms of their influence on snowmelt timing (Fig. 1)[30]. Indeed when snow is melting later in the spring (e.g., high elevation) more incoming shortwave energy is available, enhancing the radiative impact of LAPs. However, these altitudinal variations are superimposed with regional contrasts in the controlling factors of SMOD changes. The BC and dust deposition leads to larger effect on SCD in the Alps compared to the Pyrenees directly related to higher BC deposition in the Alps (Supplementary Fig. S4). The relative impact of dust compared to the one of BC is higher for the South Pyrenees than for the Alps. This is explained by the regional patterns of African dust deposition that decreases with the distance to the North African coasts according to a south-north gradient[34] and by the general predominance of African over local sources in the total dust load[38]. We can thus expect similarly large regional contrast in the impact of LAPs on snow cover in other mountainous regions.

Strong regional and altitudinal contrasts in BC and dust deposition has also been found for other mountain ranges as recently evidenced in High Mountain Asia[23]. They showed a prevailing of dust effect on snowmelt at high elevation, especially in the Western regions. The BC and dust load is about twice higher in these areas than in the Alps and the Pyrenees. The SCD shift induced by BC and dust in High Mountain Asia is thus likely higher than the 17 days reported in our study for the French Alps and Pyrenees.

Similarly strong regional contrasts arise in the trends of BC and dust deposition. Negative trends in BC deposition has been observed since the 1980s in Europe but also in North America and in the Arctic[39,40]. In contrast, positive trends have been reported in Asia, Oceanic regions and Antarctica[39]. Such positive trends may result in enhancing the effect of global warming on SCD. Regarding dust deposition, no trend is evidenced for most of the mountain regions, partly due to the strong annual variability of dust deposition and the limited record duration[25]. However, a positive trend in deposition was observed in the Caucasus likely due to increasingly dry conditions in North Africa and the middle East[41]. As exemplified in our study, these regional contrasts in deposition trend modulates the regional response of the snow cover to climate change.

## Limitations

Atmospheric and snow cover trends studied here originate from a single land surface mode driven by a single one meteorological reanalysis (S2M). However, the snow cover trend from our simulations accounting for BC and dust is in line with the observed trends in Alps and Pyrenees[7,42–44] (see "Methods" section). Besides, the results of our study are mostly based on relative differences between simulations and not on absolute values, therefore mitigating the impact of uncertainties in the meteorological variables and in the surface model. The trend in BC deposition originating from one atmospheric model reported in this study is in agreement with the negative trend observed in Europe since the 80's, as reported in many studies[19,45,46], and in total conformity with the trend in BC deposition originating from the atmospheric model GFDL-AM4[47] (Supplementary Fig. S4).

In this study, we consider only two types of LAPs, BC and dust, while other types may also contribute to modify the radiative forcing on the snow cover. Brown carbon is excluded considering the high uncertainty in atmospheric concentrations and optical properties estimates[12]. Snow algae play an important role in the snow cover energy balance in Greenland[48] but their impact on seasonal snow in the European Alps is not yet bounded and remains limited to the very end of the snow season at high elevations (typically above 2000 m a.s.l.). BC and dust are thus assumed to hold the dominant role in accelerating snow melt in the studied mountains. BC and dust optical properties are reported to be strongly variable[49–51]. The absorption efficiencies values selected here are intermediate values between extremes found in the literature and were evaluated at one location in the French Alps[33]. The sensitivity to dust refractive index is investigated using very high and low values for absorption from[51] (see Supplementary Fig. S9 and "Methods" section) resulting on a maximum uncertainty of ±5 days for ΔSMOD. The additional uncertainty associated to the choice of a scavenging coefficient for BC and dust is estimated to 1 to 2 days (see "Methods" section).

The uncertainties on the results also originate from the choice of a given snow model[52]. This is quantified using an ensemble approach with 35 configurations of the Crocus snow model. The deterministic simulation gives a slightly lower estimate of the impact of BC and dust on SMOD compared to the ensemble simulation (Supplementary Fig. S10). However, both estimates are close and the temporal trend is conserved. Finally, the impact of BC and dust deposition on snow is estimated from offline simulations and thus does not include neither the aerosol-atmosphere interactions nor the coupling between the surface and the atmosphere. Aerosols might cool the atmosphere by scattering the solar radiation before being deposited onto the surface, while snow cover reduction can lead to positive feedback enhancing the atmospheric warming by advection of turbulent heat[53] (see "Method limitations" section). Further research is required to investigate the simultaneous responses of the snow cover to these processes especially the potentially compensating effect of the particles in the atmosphere. In this study, we thus estimated that the interval of confidence associated with the average SMOD changes attributed to BC and dust is ±6 days. This confidence interval represents the uncertainty (standard deviation) due to model parameters (refractive index, scavenging coefficient and model physics) as detailed above and in the "Methods" section.

## Conclusion

Our results demonstrate that BC and dust deposition advances the end of the snow season by 17 ± 6 days on average over the French Alps and the Pyrenees mountain ranges and the peak snowmelt runoff by 2 weeks. Given the relevance of the snow cover in terms of water resources and tourism in these regions, both effects have deep socio-economic implications. It is also a significant effect to consider in mountain ecosystems studies. The BC effect is likely prevailing over

the dust effect in the North Alps. BC deposition alone shortens the snow season by 11 days on average, compared to pure snow simulations. However, the BC effect would be even larger without the decrease in BC deposition observed since the 1980s. At the beginning of the study period (1979–1988), a shortening of 15 days is attributed to BC deposition alone. On the contrary, for the recent period (2009–2018), the shortening due to BC is of 10 days only on average. This emphasizes the key effect of changes in BC deposition on the snow cover. Due to the current warming, the snow cover decline would be even more pronounced without the opposing effect of the decrease in BC deposition since the 1980s. The future changes in BC deposition related to human activities will therefore be of critical importance for the evolution of the snow cover in the coming decades. Hence, past and future trends in snow cover and LAPs deposition cannot be investigated independently. This is crucial to assess the future regional response of mountain hydrology, water availability, and the future evolution of mountain social-ecological systems.

## Methods

### Site description and atmospheric forcing
**Study sites: French Alps and Pyrenees.** The study covers the French Alps and the Pyrenees mountain ranges. In the French Alps, most of the mountains range between 300 and 3600 m a.s.l. with a maximum at the Mont-Blanc with 4810 m a.s.l.. In this study, the Alps are split into the North Alps, with elevation ranging between 1500 to 3000 m a.s.l. and the South Alps with elevation ranging between 1800 to 3000 m a.s.l.. These elevation ranges are selected as the ones fulfilling the condition of a mean simulated snow depth >30 cm over the winter period (i.e., 1st of December to 30 of April), and considering all the winters over the 1979–2018 period. Elevation above 3000 m a.s.l. were excluded because the meteorological forcing and the snow model Crocus were not thoroughly evaluated int this context (see details below).

The Pyrenees mountain range covers distinct mountains located in France, Spain, and Andorra, and most of them range between 300 and 3000 m a.s.l. with a maximum of 3404 m.a.s.l at the Aneto Peak (Spain). In this study, the Pyrenees are divided between the North and the South Pyrenees. The elevations selected, fulfilling the snow depth condition (i.e., an average larger than 30 cm for the winter period) are from 1800 to 3000 m a.s.l for the North Pyrenees and from 2100 to 3000 m a.s.l. for the South Pyrenees.

The French Alps correspond to the headwaters of the Rhône river, one of the major rivers in Europe, and many of its tributaries; the Pyrenees correspond to the headwaters of several major rivers in southern France (Garonne and Adour rivers) and northern Spain (Ebro river). The French Alps and Pyrenees experience different climatic conditions, spanning the Mediterranean to continental climates in Europe. The French Alps and Pyrenees host the vast majority of the ski resorts in France. French ski industry is ranked third globally in terms of skier visits over the past decades. The changes in snow cover melt-out date spanning the French Alps and Pyrenees is thus of substantial relevance for the national ski industry. This case study thus provides a basis for further studies carried out at the European scale or addressing other mountain regions in the future.

**S2M meteorological reanalysis.** The S2M atmospheric reanalysis is built from the Système d'Analyse Fournissant des Renseignements Atmosphériques à la Neige (SAFRAN) meteorological analysis system[28]. SAFRAN data are based on atmospheric vertical profiles simulated by an atmospheric model (ERA-40 reanalysis until 2001 and ARPEGE numerical weather prediction model after 2002). These guesses are then corrected by optimal interpolation with mainly observed surface meteorological data from various networks (automatic or manual observations). SAFRAN do not assimilate any observation above 3000 m a.s.l..

SAFRAN outputs include hourly meteorological variables: 2 m air temperature and relative humidity, precipitation amounts and phases, incoming direct and diffuse shortwave radiation, incoming longwave radiation, wind speed, cloudiness. These data are assumed to be homogeneous within a given "massif". A massif is a conceptual object corresponding to a mountainous area (of about 1000 km² on average) over which the meteorological conditions are considered homogeneous at a given elevation. In this geometry, the Alps and the Pyrenees are both divided into 23 massifs (Fig. S11) selected for their climatological homogeneity[54]. The S2M reanalysis uses a 300 m vertical resolution[28]. In our study, for each massif, meteorological data depends only on elevation (one data point every 300 m) considering a flat surface.

**BC and dust deposition from CNRM-ALADIN63.** BC and dust deposition fluxes are obtained from the regional climate model CNRM-ALADIN63[29]. This model includes an interactive tropospheric aerosol scheme able to represent the main aerosol species such as BC and dust in the troposphere. These aerosols are prognostic variables, subject to transport, dry deposition and in-cloud and below-cloud scavenging. In this study, hourly output of dry and wet BC and dust deposition fluxes were used, coming from a simulation over the 1979–2018 period driven by the ERA-Interim reanalysis in order to ensure a realistic timeline of the evolution of aerosol deposition. This simulations was carried out on a regional domain covering Europe, the Mediterranean Sea and northern Africa, at a 12 km horizontal resolution with 91 vertical levels. Data from this simulation have been chosen as the study performed by ref. 33 indicated a good agreement with the observations at the Col du Lautaret (Alps, France).

Points covering the Alps and Pyrenees are extracted from this simulation, for the period 1979–2018, and downscaled to obtain BC and dust forcing in SAFRAN geometry. For that purpose, for a given massif, points located within the massif boundaries were selected. Between 5 to 10 points were selected, depending on the massif area. From these selected points, the hourly mean and the hourly altitudinal gradient of each variable: dry and wet BC and dust deposition is computed. The hourly gradient is then applied to the hourly mean to redistribute the data for each 300 m steps in agreement with SAFRAN geometry. Due to differences in precipitation timing between SAFRAN and CNRM-ALADIN63, and to agree with SAFRAN atmospheric forcing data, hourly dry and wet deposition fluxes are added and redistributed according to SAFRAN precipitation. Uncertainties of such approach are evaluated (see "Method limitations" section).

### Snow cover simulations and evaluation
**Simulations description.** Simulations are performed with the detailed multilayer snowpack model SURFEX/ISBA-Crocus[55]. Crocus is a detailed snowpack model that includes, among others, dynamical layering, full description of the surface energy balance, specific surface area, liquid water content as prognostic variable, snow types, dry and wet snow metamorphism, and an explicit representation of LAPs (the number and types are users defined) with a fully coupled spectral radiative scheme for solar radiation absorption inside the snowpack at 20 nm spectral resolution[27]. The radiative properties of dust and BC can be user-defined either based on LAP refractive index or on LAP mass absorption efficiency[27,33]. In the present study, dust is defined by the mass absorption efficiency from ref. 51 corresponding to dust PM2.5 from Libya in agreement with the study performed by ref. 33. BC is defined from the constant refractive index advised by ref. 49 (i.e., $m = 1.91 - 0.79i$). The MAE is then scaled to obtain an MAE value at 550 nm of $11.25\,m^2\,g^{-1}$[56]. The scaling makes it possible to implicitly account for the potential absorption enhancement due to internal particle mixing or particle coating.

The model ran over the period 1979–2018, in a semi-distributed geometry (i.e., per 300 m elevation bands, for each massif, following

the SAFRAN geometry). For each simulation point, the snow cover fraction is set to 1 whenever snow is present on the ground. Note that only seasonal snow is simulated. In our simulations we are not accounting for snow on glacier. LAPs accumulated during one season are thus removed with runoff when the snow cover melts out. Contrary to some other applications of the SAFRAN-Crocus system, we did not extend the simulations to different aspects and slopes. To investigate the explicit effect of BC and dust on snow cover evolution, four configurations are set up: (1) pure snow simulation with no LAP ($S_{pure}$), (2) simulation considering BC and the dust deposition fluxes ($S_{BC+Dust}$), (3) simulation considering BC deposition only ($S_{BC}$) and (4) a simulation considering dust deposition only ($S_{Dust}$). Furthermore, the original version of Crocus model[55] considers an implicit representation of LAPs. Indeed, the snow albedo decreases based on the snow age and therefore implicitly considers a darkening of the snow surface. However this representation is not able to consider temporal variability of the deposition. For model evaluation, simulations ($S_{baseline}$) performed with this version are compared to the one allowing an explicit representation of BC and dust (i.e., $S_{BC+Dust}$).

**Evaluation.**
(1) Using satellite images
   For each massif, the simulated SCA and the SCD are compared to the MODIS snow product. MOD10A1 (Terra) and MYD10A1 (Aqua) snow products are downloaded from the National Snow and Ice Data Center[57,58] for the period 2000–2016. The binary snow products are projected on a 500 m resolution grid in the same coordinate system as the DEM. Missing values, mainly due to cloud obstruction, are interpolated using a stepwise gapfilling algorithm that was evaluated in the Pyrenees[59]. The simulated snow depth is projected on the same 500 m DEM resolution. Then, a threshold of 0.03 m is used to convert the simulated snow depth into snow presence or absence for each grid cell[59]). By comparing the simulated and the observed SCA, results indicate high correlations ($R^2 > 0.75$) and low RMSE (i.e. <25%) for all the simulations. Considering the explicit representation of BC and dust generally leads to a lower RMSE and a lower bias compared to the other simulations (Supplementary text, Supplementary Fig. S11). Indeed, for the simulation considering BC and dust, the RMSE is ranging between 9% and 18.3% depending on the massif, with a mean of 12.3% and 13.8% forth the North and South Alps and 13.5% and 15.9% for the North and the South Pyrenees respectively.
(2) Comparison with snow depth measurements
   The simulations are evaluated using 495 snow depth measurements stations located in the French Alps and Pyrenees, spanning an elevation range from 1200 to 2700 m a.s.l., over the period 1983–2018. Two types of evaluation are carried out. First, we compare simulated and observed SMOD (Supplementary Table S1). Three error metrics were used: the root mean square error (RMSE), the mean absolute error (MAE) and the mean bias (a positive bias indicates a later simulated SMOD compared to the observation). Results show that considering the explicit representation of BC and dust outperforms the pure snow simulation set-up with an overall bias of −0.68 days, a RMSE of 19.2 days and a MAE of 13.3 days. The RMSE is influenced by extreme values, frequently due to one snow free day in early winter in the observed time series. The RMSE calculated only for SMOD later than March 1st drops to 15.7 days (MAE 11.4 days). In addition, model evaluation using observed meteorological forcing at the station scale[27,30,33] shows that the SMOD simulations is accurate within ±1–2 days. Most of the RMSE in Table S1 likely comes from the spatial scale discrepancy between the meteorological forcing and snow observations, and possibly other residual errors in the meteorological forcing. To investigate the statistical significance

of the differences in scores between $S_{pure}$ and $S_{BC+Dust}$ simulations, we use a bootstrap experiment. Namely, we randomly draw 35 years with replacement over the 1983–2018 period, which includes most of the in situ observations. We therefore obtain 35 random years, and as the draw is with replacement, years can be drawn several time, or not drawn. The random draw is performed 1000 times for both $S_{pure}$ and $S_{BC+Dust}$ simulations, in order to obtain 1000 series of 35 years, providing 1000 values of bias, RMSE and MAE. The bias, RMSE and MAE distributions show that the difference in bias, RMSE and MAE between $S_{pure}$ and $S_{BC+Dust}$ simulations are statistically significant. This implies that the ΔSMOD attributed to BC and dust is greater than the uncertainties of the model itself. Since SMOD is strongly affected by uncertainties in the meteorological forcing, an additional evaluation is carried out focusing on daily melt rates. The evaluation is done by comparing the simulated and measured daily variations of snow depth in case of melt (ref. 60, Supplementary text). Considering the explicit representation of BC and dust generally leads to a lower bias compared to the other simulations (Supplementary Fig. S12).

**Strategy of BC and dust contribution quantification.** The BC and dust contribution is quantified at a regional scale, considering four regions: North and South Alps and Pyrenees. The choice to work at a regional scale limits the uncertainties of SAFRAN reanalysis compared to the uncertainties estimated at the local scale of a station that are mainly impacted by local effects.
(1) On the snow cover duration
   First, the SMOD, defined as the last date of the longest period with at least 30 cm of snow is computed for each simulation. Then, to quantify the impact of BC and dust on the shortening of the season, the SMOD differences between $S_{pure}$ and $S_{BC+Dust}$ are computed for each massif, elevations, and years. The differences (i.e., ΔSMOD) are given by elevation and locations (Fig. 1), and the spread corresponds to the annual variability. The differences are also quantified for the simulation between $S_{pure}$ and $S_{BC}$ and $S_{Dust}$ to evaluate the influence of BC and dust deposition on the SMOD separately (Fig. 1) Finally, differences between $S_{pure}$ and $S_{baseline}$ are also shown.
(2) On the annual variability of the snow cover
   A statistical approached based on the variance analysis (ANOVA) is used to evaluate the contribution of the BC and dust on the SMOD annual variability. The detailed method is described in ref. 32 (following the Eq. reported in the Supplementary text). Here, the contribution of the two parameters "meteorological conditions" and "BC + Dust" to the SMOD variance is computed (Fig. 2).
(3) On the trend in snow cover
   The trends in meteorological forcing (temperature and solid precipitation), BC and dust, are computed over the 1979–2018 period, and a Student's test (t test with a 0.05 confidence interval) is applied to evaluate the significance. The trends in simulated snow cover (SMOD) and CMD of $S_{pure}$ and $S_{BC+Dust}$ are quantified following the same method (Fig. 3). Then, for each year, the SMOD differences between $S_{pure}$ and $S_{BC+Dust}$ (and also $S_{BC}$ and $S_{Dust}$ separately) is computed, in order to obtain a dataset (39 years) from these differences. The trend of this dataset is then computed and its significance is evaluated following the t-test with a 0.05 confidence interval. This method allows to evaluate the significance of the impact of the BC and dust on the trend.

**Hydrological impacts.** To evaluate the impact of BC and dust on the hydrology, the CMD of the runoff from snowmelt is computed for each simulation. The CMD is defined as the date for which 50% of the total annual runoff from snowmelt is reached[35]. The simulated runoff from snowmelt by Crocus is projected on a 500 m DEM for the Alps and the

Pyrenees. The North and South Alps and the Pyrenees are considered as four distinct catchments and the total runoff is computed for each catchment for the 39 years. The mean with the standard deviation (indicating to the temporal variability) is computed for each catchment considering $S_{pure}$ and $S_{BC+Dust}$, to evaluate the impact of the BC and dust on the runoff amount and timing (Fig. 4). Then, the impact of BC and dust on the annual variability of the CMD is evaluated following the same method as for the snow cover described above (Supplementary Fig. S8). The same method as for the snow cover is also applied to evaluate the impact of the BC and dust on the CMD trends (Fig. 3).

### Method limitations
#### Forcing uncertainties.
(1) Trends

As atmospheric and snow cover trends studied here, originate from only one surface model forced with only one meteorological reanalysis (S2M), comparisons to other models and observations are required to reinforce our conclusions. All climatological reconstructions are affected by irreducible uncertainties in the obtained trends due to the temporal heterogeneities of the number and quality of the data upon which they rely. In particular, ref. 61 demonstrate that the heterogeneities of available temperature measurements can significantly affect local temperature trends in S2M. This limitation is expected to be less important when considering large scale signals as in this paper. However, ref. 62 and ref. 63 revealed some large scale differences in temperature and precipitation trends between S2M analyses and the Regional Climate Model, MAR, forced by ERA20C reanalysis at the boundaries but not assimilating any observation inside the simulation domain. Similar discrepancies were obtained in Switzerland by ref. 64 comparing similar products between S2M and another atmospheric model. However, the snow cover trend from our simulations accounting for BC and dust is in line with the observed trends. In our simulations, the SMOD trend in the North Alps at around 1500 m a.s.l is $-4.0 \pm 0.6$ days per decade, comparable to the trend observed at Col de Porte ($-4.1$ days per decade, 1325 m a.s.l., 1960–2018)[42]. In addition, the averaged trend of $-5.3 \pm 0.6$ days per decade (3.0%) simulated for the North Alps considering elevation ranging between 1500 and 2400 m a.s.l. is in good agreement with an averaged earlier snowmelt of 5.7 days per decade observed at 11 stations (covering elevations from 1139 to 2540 m a.s.l) in the Swiss Alps over the 1970–2015 period[43]. In a study based on 202 to 688 stations in the European Alps over the 1971–2019 period, the SCD computed over the March–May season decreases by $-7.8$ to $-0.7$% depending on elevation (1000–3000 m a.s.l.) and orientation (North or South)[7]. The mean relative change of 3.0% from our simulations is within the range of this study. In the Pyrenees, the absence of significant trend of the SMOD at 2100 m a.s.l. since 1980 is in agreement with the work of ref. 44. Besides, the results of our study are mostly based on relative differences between simulations and not on absolute value, therefore mitigating the impact of uncertainties in the meteorological variables and in the surface model.

(2) SAFRAN reanalysis

In snow modeling, most of uncertainties are brought by the meteorological forcing[65]. Uncertainties in the SAFRAN meteorological reanalysis were assessed in details in ref. 28. The uncertainties are variable in time and space. Ref. 60 reported a bias of the shortwave radiation at some locations in the Pyrenees. Ref. 66 showed an underestimation of precipitation amount in the Alps at the highest elevations. Ref. 28 showed that the temperature accuracy varies between 1 and 1.5 K and the accuracy of the monthly cumulated precipitation is roughly

20 kg m$^{-2}$ with a low bias of less than 2 kg m$^{-2}$ at 43 stations. However, in our study, the impact of meteorological input uncertainty is minimized as our conclusions are based on relative differences between simulations with identical meteorological forcing except LAP deposition and not on absolute values.

(3) BC and dust

BC and dust deposition used to force the snowpack model is from only one atmospheric model and could be taken with caution. Still, BC and dust deposition from CNRM-ALADIN63 have been evaluated by ref. 33 at the Col du Lautaret (French Alps) over two snow seasons. Using in situ spectral reflectances, they evaluated the radiative impact of BC and dust simulated with Crocus using CNRM-ALADIN63 model deposition fluxes as inputs. They reported an $R^2$ of 0.78 and mean bias in BC equivalent surface concentration of 31 ng g$^{-1}$ results slightly underestimate the observed values). Such an evaluation at one location is not sufficient. In our study, we indirectly evaluated the simulated radiative impact of BC and dust by evaluating SMOD and snowmelt rates using 495 snow depth measurement stations (Table S1 and Fig. S12). For both SMOD and snowmelt rates, the simulations using BC and dust from CNRM-ALADIN63 outperform the other simulations. This provides further confidence in the overall accuracy of the simulated radiative impact, i.e., the combination of the deposition fluxes and the optical properties of BC and dust (see Model uncertainties). Figure S4 compares the annual BC and dust deposition values from CNRM-ALADIN63 and from the global model GFDL-AM4[47,67]. BC deposition and trends values from CNRM-ALADIN63 are in agreement with BC deposition modeled by GFDL-AM4 (mean bias of 0.004 g m$^{-2}$, corresponding to about 9% of the mean BC loads) while dust loads from CNRM-ALADIN63 are higher by a mean factor of 3.7 compared to the ones from GFDL-AM4. This might be related to the coarser spatial resolution of GFDL-AM4. Both models indicate no significant trend in dust deposition over the period 1980–2014.

The downscaling method for dust and BC deposition fluxes is also a source of uncertainties. We tested the impact of the number of points selected in each massif to compute the gradient of deposition fluxes. Results shows that differences in total BC and dust deposition are lower than 5%. The methodology to redistribute dry and wet deposition according to SAFRAN precipitation might also lead to uncertainties. We performed of sensitivity test with two extreme scenarios (i) all the fluxes are considered as dry deposition, (ii) all the fluxes are considered as wet deposition. Scenario (i) leads to a mean difference of 1.8 ($\pm 1.4$) days with the simulations used in the study. For scenario (ii), the mean difference is $-10$ ($\pm 2.3$) days. These scenarios provide the uppermost and lowermost boundaries of the uncertainties due to this dry/wet partitioning. Even in such extreme cases, the difference is lower than the 17 days of shortening attributed to dust and BC.

#### Model uncertainties.
(1) Optical properties of BC and dust

The choice of dust refractive index and mass absorption coefficients also implies some uncertainties. In this study, values for Saharan Libya PM2-5[51] are selected: a mass absorption efficiency at 400 nm of 110 10$^{-3}$ m$^2$ g$^{-1}$ and a dust Angström exponent equal to 4.1. This choice is made as Saharan dust is the primary source of dust deposition in European mountainous areas and as ref. 33 demonstrated based on spectral reflectance measurements that this spectral signature agrees well with the measured spectrum for two winter seasons. However, this spectral signature is expected to vary with the source location which varies also over time. Simulations performed by changing this spectral signature,

with a mass absorption efficiency at 400 nm ranging between 27 $10^{-3}$ m$^2$ g$^{-1}$ (corresponding to the source Sahel–Bodélé PM10) and 630 $10^{-3}$ m$^2$ g$^{-1}$ (for dust PM2-5 from Sahel–Mali). The Angström exponent is equal to 3.3 (Bodélé) and 3.4 (Mali). These two indices are selected as they represent low and high values absorption for African dust[51]. This sensitivity study allows quantifying the uncertainty related to the spectral signature chosen (Supplementary Fig. S9). Depending on this parameterization, the median of the ΔSMOD at 2100 m a.s.l. varies between 15 to 25 and 12 to 23 for the North and South Alps respectively, and between 11 to 20 and 10 to 21 for the North and South Pyrenees respectively. The median of the ΔSMOD with the spectral signature chosen in this study is closer to the lower median (i.e., 19 (15) for the North (South) Alps and 13 for both the North and South Pyrenees). This suggests that the effect of dust reported in this study might be under-estimated for Saharan dust deposition events originating from locations close to Mali. Finally, uncertainties are also associated with the modeling of BC absorption efficiency in snow. Indeed, the evolution of this variable is still poorly understood with variations of at least a factor 2 reported in the literature (e.g., ref. 50 and references therein). We thus estimate that the uncertainties associated with the selected optical properties of LAPs range within ±5 days for Δ SMOD (Fig. S9).

(2)  Scavenging of BC and dust

In this study the scavenging of impurities in the snowpack was disabled as in ref. 33. Indeed, the lack of quantitative observations of LAPs percolation at the snow surface in presence of melt water does not allow a proper evaluation of this effect. A sensitivity test to the scavenging coefficient was performed allowing 20% of the BC (5% of the dust) to be scavenged with the water percolation[27]. This leads to a small effect on the value of Δ SMOD (i.e., a median of the ΔSMOD variation at 2100 m a.s.l. of 2 days for the North Alps and 1 day for the South Alps).

(3)  Representation in snow physics

Large uncertainties remain also in the representation of the snow physics, and ref. 68 quantified a SMOD difference around 30 days depending on the complexity of the snow model chosen (degree-day snow model *vs.* detailed snow model). Using an ensemble approach considering 35 different state-of-the art parameterizations in Crocus, ref. 30 indicates a SMOD variation around 5 days, highlighting the uncertainty related to the selected snow physic in the model. In this study, we quantified the model uncertainty using and ensemble approach considering 35 different state-of-the art parameterizations in Crocus[52] to simulate the ΔSMOD at 2100 m a.s.l. (Supplementary Fig. S10). Uncertainties related to model physic choice and location (i.e., box plot size of the ensemble simulations) is within the same range than the location uncertainty (i.e., the box plot size of the deterministic simulation in Supplementary Fig. S10).

(4)  Geometry of the simulations

Simulations are performed in a semi-distributed geometry only considering a flat aspect. This is because the slope and aspect represent a huge amount of additional data, complex to be considered in the simulations. Still, with a simulation considering the slope and aspect, we expect a higher impact of the BC and dust deposition on the snowpack evolution for southern slopes compared to flat areas (and the opposite for the northern slopes). Southern slopes are indeed exposed to considerably higher solar radiation.

(5)  Offline simulations

Simulations are performed in offline mode, i.e., the aerosol-atmosphere interactions and the coupling between the surface and the atmosphere are not considered. The enhanced melt estimated in this study could be partly compensated by the cooling effect of the aerosols in the atmosphere before their deposition. Indeed, for instance, ref. 69 quantified an enhanced melt due to BC of 3–12 kg m$^{-2}$ while the aerosol-radiation interactions dampen the surface of incoming solar energy to a level equivalent to the preservation of 1–5 kg m$^{-2}$ of snow. Ref. 70 showed an effect of similar magnitude of radiative warming in the snowpack and the magnitude of surface radiative cooling due to BC and dust in the atmosphere (10 W m$^{-2}$). On the opposite, not accounting for the coupling between the surface and the atmosphere coupling can result in an underestimation of the aerosol effects on snow[71,72]. Further research is warranted to investigate the above-mentioned feedbacks at a regional scale in mountainous areas.

## Data availability
The dataset and the files required to reproduce the simulations presented in this study are available at https://doi.org/10.5281/zenodo.6760050. The S2M reanalysis is available at https://doi.org/10.25326/37#v2020.2.

## Code availability
TARTES model is available here: http://snowtartes.pythonanywhere.com/ (web application and Python module download). The Crocus snowpack model (including all physical options of the ESCROC system) is developed inside the opensource SURFEX project. The SURFEX version used in this work is the same as in Cluzet et al.[73] (git tag CrocO_v1.0; source code available at https://doi.org/10.5281/zenodo.3774861, see this reference for more detailed technical information about code access).

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

## Acknowledgements

The authors thank Florent Dominé, Matthieu Vernay and Delphine Six for helpful conversations. We also thank the scientific editor Kasey Bolle sand two anonymous reviewers for their comments and suggestions, which significantly improved the quality of the manuscript. CNRM/CEN and IGE are part of Labex OSUG@2020 (investissement d'avenir—ANR10 LABX56). M.D. has received funding from the European Research Council (ERC) under the European Union's Horizon 2020 research and innovation programme (grant agreement no. 949516, IVORI). This study was supported by the ANR programme ANR-16-CE01-0006 EBONI; and the CNES APR grant MIOSOTIS. This research was at least partially supported by Lautaret Garden-UMS 3370 (Univ. Grenoble Alpes, CNRS, SAJF, 38000 Grenoble, France), member of AnaEE-France (ANR-11-INBS-0001AnaEE-Services, Investissements d'Avenir frame) and of the eLTER-Europe network (Univ. Grenoble Alpes, CNRS, LTSER Zone Atelier Alpes, 38000 Grenoble, France). As part of the project CDP TRAJECTORIES, M.M. has been funded by the French National Research Agency in the framework of the "Investissements d'avenir" programme (ANR-15-IDEX-02). P.G. received support from the NASA Understanding Changes in High Mountain Asia (NNH19ZDA001N-HMA) and the Earth Surface Mineral Dust Source Investigation (EMIT), a NASA Earth Ventures-Instrument (EVI-4) Mission.

## Author contributions

M.R. led the study and performed the simulations and analysis. M.D. designed the study. M.R. and M.D. wrote the paper with contributions from all co-authors. M.D. revised the paper with the help of all co-authors. S.G. provided the MODIS data and contributed to the analysis. F.T., M.L. and R.N. implemented the Crocus version with explicit representation of LAPs. P.N. and P.G. provided atmospheric simulations including LAPs deposition and contributed to the analysis. A.R. contributed the ANOVA and trends computations and analysis. M.M. contributed to the downscaling method for the LAPs fluxes and to the analysis. S.M. and G.P. contributed to the analysis of the results and discussed and commented the manuscript with all the co-authors.

## Competing interests

The authors declare no competing interests.
