## [Peer Review File · Nature Communications]

Black carbon and dust alter the response of mountain snow cover under climate changeReviewers' Comments:

Reviewer #1:

Remarks to the Author:

This paper assessed the impacts of light-absorbing particles (black carbon and mineral dust) on snow cover changes and hydrology at two mountain regions using numerical simulation. Although the long-period simulation (39 years) combined with snow cover dynamic analysis provide valuable information for evaluating the responses of mountain snow cover to black carbon (BC) and dust and the environmental implications, there are some issues should be detailed addresses and improved.

The major ones are the introduction and methodology. The system discussion on the limitation and implications should also be considered carefully. The most important issue I concern is that how did the author performed the BC and dust deposition during 1979-2018 by using regional climate model. Snow cover is different from glacier. Snow cover will melt and disappeared in summer season in these two mountain regions, which will not accumulate BC and dust deposition. From Figure S2, the author only showed the annual BC and dust deposition, I guess the author use the annual deposition of BC and dust and evaluate the impact on snow cover melting? How did the author input the BC and dust deposition date to the snow cover in the model?

For the current title, what kind of response of snow cover to black carbon and dust? I see from the results, the author wants to show BC and dust advanced the snow melting and runoff peak. The title should be revised.

For the abstract, no results on the hydrological impact by BC and dust, which is the most important part in this study.

Other concerns are as below:

1. For the introduction, the brief review on research progress of BC and dust and their climate effect on snow melting was not completed (see below references). Especially in Arctic, North America, and High Mountain Asia, the research results and gaps were not mentioned and summarized, which limited this study to abroad audience from a global view of this topic. Meanwhile, no description on the importance of the two regions- Alps and Pyrenees mountains, why you choose these two regions? At the end of the introduction, no specific purpose of this study was delivered. The reason why the author excluded the discussion on brown carbon and algae was not sufficient.

Kang S., Y. Zhang, Y. Qian, H. Wang. 2020. A review of black carbon in snow and ice and its impacts on cryospheric change. *Earth-Science Reviews*, 210, 103346.

<https://doi.org/10.1016/j.earscirev.2020.103346>.

Kang S., Q. Zhang, Y. Qian, Z. Ji, C. Li, Z. Cong, Y. Zhang, J. Guo, W. Du, J. Huang, Q. You, A. K. Panday, M. Rupakheti, D. Chen, Örjan Gustafsson, M. H. Thiemens, D. Qin. 2019. Linking Atmospheric Pollution to Cryospheric Change in the Third Pole Region: Current Progresses and Future Prospects. *National Science Review*, 6(4): 796-809. <https://doi.org/10.1093/nsr/nwz031>.

Schmale, J., Flanner, M., Kang, S., et al., 2017. Modulation of snow reflectance and snowmelt from Central Asian glaciers by anthropogenic black carbon. *Sci. Rep.*, 7:40501. DOI: 10.1038/srep40501

Zhang Y. L., S. Kang, M. Sprenger, Z. Cong, T. Gao, C. Li, S. Tao, X. Li, X. Zhong, M. Xu, W. Meng, B. Neupane, X. Qin, M. Sillanpää. 2018. Black carbon and mineral dust in snow cover on the Tibetan Plateau. *The Cryosphere*, 12: 413-431. <https://doi.org/10.5194/tc-12-413-2018>.

2. The snow cover area and snow depth simulations were evaluated using satellite data and in situ observations. BC and dust deposition flux data used for model simulation also should be evaluated or the uncertainties should be addressed. It seems like the dust deposition used in this study was overestimated compared with other study in Figure S2.

3. In line 76, "Shortening of the season due to BC and dust" is not clear, which season?

4. Lines 82-84, the logic of this sentence is not right. From this sentence and Figure 1, No results indicated the variations of BC and dust reduced SMOD.

5. Lines 93-95, the sentence is not completed. Also simulated results show a larger absolute impact of BC and dust on the SMOD at higher elevations, and authors explain this is associated with later melt and longer snow season at higher elevations, where dust and BC amplify its effect on snowmelt due to exposed to higher incoming solar radiation. Is this mechanism can be revealed in this snowpack model

or only speculation?

6. Figure 2 presented the parameters contribution to SMOD (snow melt-out date), including weather, BC, and dust. Here, using "climate" is more appropriate than "weather".

7. In the second part of the results (Figure 2), the contribution of meteorological condition to SMOD was much larger than BC and dust. The importance of BC and dust on SMOD in these two mountain regions need be clarified further.

8. The impacts of BC and dust on snow cover changes by analyzing SMOD changes were quantified only. Other parameters, such as snow surface albedo changes, can be a directly evidence for quantifying the impacts of light-absorbing particles on snow cover and should be discussed.

9. The statements in lines 111-119 were not clear and should be strengthened. It seems like that aged snow related to meteorological conditions can lead to the enrichment of light-absorbing particles and had stronger impacts on snow cover, and the contribution of this effect was separated and analyzed in this part? And how to separate the contribution of BC and dust only without considering this effect?

10. Line 128, What is "atmospheric trends"?

11. Discussion on the "past trends", in the result of Shortening the season due to BC and dust, the Figure 1 showed the impact of BC and dust on SMOD was larger in the high elevations (>3000m), then why discussed the results at 2100 m asl?

12. Lines 145-146, the sentence is difficult to understand. When BC and dust deposition to the pure snow, BC will enhance the melting rate. If less BC deposition, the melting rate became smaller, but still enhance the melting rate compared to pure snow, rather than "decrease melting rate". You can check the reference Flanner et al., 2007; Skiles et al., 2018.

13. Lines 147-148, the dust did not significantly affect the SMOD trend. Why the dust affected the SMOD, then had almost no effect on SMOD trend? Please explain more on this sentence. The Figure3 and Figure S2 is kind of paradox. Make them clearer.

14. It was concluded that the effect of BC deposition on SMOD was prevailing over the dust effect. But the uncertainties due to mass absorption efficiency, deposition flux, snow cover fraction, scavenging efficiency in snow were discussed relatively less. Those are fundamental settings to models of radiative transfer and could cause large biases.

15. The SURFEX/ISBA-Crocus snowpack model is driven by the meteorological reanalysis S2M, whereas the BC and dust deposition fluxes from the CNRM-ALADIN63 model is driven by the ERA-Interim reanalysis. How do you evaluate the impact of uncertainties in using different meteorological reanalysis datasets to quantify respectively the deposition fluxes of BC and dust and its effects on snow cover change?

16. The meteorological reanalysis S2M (one data point every 300 m) is used to drive the snowpack model. What is the spatial resolution of the SURFEX/ISBA-Crocus simulation here? How do you avoid the biases due to large resolution differences between different datasets used in the snowpack model, i.e., S2M (300m), BC and dust deposition flexes from the regional climate model CNRM-ALADIN63 (12km)?

17. Authors only show the simulated results on average in the North and South Pyrenees (left) and the North and South Alps, respectively. It is better to display the results on the grid points, helping readers to understand the spatial heterogeneity of the effects of BC and dust on snow cover change and its hydrological implications.

18. The authors discuss in detail the uncertainties of the driving datasets as well as parameters in the snowpack model. So many uncertainties may result in big simulation biases. Can you quantitatively assess these uncertainties? For example, previous study reported the S2M reanalysis underestimated of precipitation amount in the Alps. Meanwhile, the SMOD decreases are less than 3% of the snow season duration due to BC deposition in Line 142. If the decreased SMOD due the underestimated precipitation is greater than that from BC deposition, these simulation evaluations will be not enough to convince the readers. Thus, I suggest authors firstly quantified the uncertainties at the observation station. If the biases caused by the drivers and model parameters is acceptable, you then start to simulate in regional scale. It is better for authors to provide confidence interval in quantifying the effect of BC and dust on snow cover, to show the uncertainties of the simulations.

19. As for impacts on mountain hydrology, no significant change in the intensity of the peak runoff is

attributed to the presence of BC and dust in Fig. 4. Can you explain the reasons accounting for this phenomenon, considering BC and dust lead to a shift in the CMD with earlier snowmelt?

Reviewer #2:

Remarks to the Author:

Assessments of long-term impact of LAPs at the regional scale are few, and this manuscript aims to address the same over French Alps and the Pyrenees over the 1979–2018 period using offline numerical simulations. Their analysis illustrates that dust and BC deposition has advanced the region's snowmelt by 17 days on average which can be a major implications for water availability. The relative influence of dust and BC in elevational gradient and spatial variations of reduction in SMOD is aptly reported. Further, they analyse the interannual variability and attempts to separate out the BC and meteorology impacts. This is a well-organized and interesting study which is advancement in the field. However, a few things are not clear.

1) The entire analysis is based on simulated SMOD, hence detailed evaluation of simulated SMOD should be included.

2) The methodology of segregation of LAP and warming effect on SMOD is not clear. More information should be given in the ANOVA analysis as understanding the analysis is hard in the present form. Actually, what is the response of SMOD to warming is not clear. The title says that LAPs alter this response. For instance, the sensitivity of SMOD to LAP can also changing with warming, how is it accounted?

3) The estimates of LAP-induced changes in SMOD or CMD is $\sim 10\%$. These estimates are from offline simulations and do not include the aerosol-atmosphere and aerosol-snow feedbacks and couplings. Authors have mentioned many aspects of meteorological uncertainty and also justified the estimations. But, a clear discussion on uncertainties due to offline mode of analysis (comparing their findings with LAP-snow coupled simulations) should be included. please see: Rahimi et al., 2020 ACP; Sarangi et al., 2019 ACP; Usha et al., 2021 ERL; Zhao et al., 2014; ACP etc

Response to reviews

Black carbon and dust alter the response of mountain snow cover under climate change

Marion Reveillet, Marie Dumont, Simon Gascoïn, Matthieu Lafaysse, Pierre Nabat, Aurelien Ribes, Rafife Nheili, Francois Tuzet, Martin Menegoz, Samuel Morin, Ghislain Picard, and Paul Ginoux

February 8, 2022

Introduction

We would like to thank the reviewers for the evaluation of our work, the encouraging feedback and comments. We have substantially revised our manuscript according to the reviewers and editor comments. The method section was revised and extended, especially the ANOVA section. The uncertainties were quantified in detail. We also provide an extended evaluation of the simulations using station measurements as requested by both reviewers. A point-by-point response to each comment is provided below with the proposed changes in the manuscript.

In the present document, comments from the referees are reported in the blue boxes before the authors response. In the response we use *italic* fonts to quote text from the manuscript and **blue** to highlight changes. A track-changes version of the revised manuscript is also provided.

Reviewer #1

General comments

This paper assessed the impacts of light-absorbing particles (black carbon and mineral dust) on snow cover changes and hydrology at two mountain regions using numerical simulation. Although the long-period simulation (39 years) combined with snow cover dynamic analysis provide valuable information for evaluating the responses of mountain snow cover to black carbon (BC) and dust and the environmental implications, there are some issues should be detailed addresses and improved. The major ones are the introduction and methodology. The system discussion on the limitation and implications should also be considered carefully.

The authors thank the reviewer for his/her reading and his/her feedback on the paper. The introduction of the paper has been revised according to specific comment 1 (see below). The methodology has been completed and revised in agreement with specific comments 9, 15, and 17. In addition, in response to specific comments 2, 14, 16 and 18, we provided more details on the uncertainties in the "Limitations" section as well as in the "Method limitations" section. We estimated the effect of BC and dust on snow albedo and absorbed energy as requested by the referee. We provided an interval of confidence for the shortening attributed to BC and dust. We included an additional evaluation of the snow melt out date (SMOD) accuracy using nearly 500 measurement stations across the French Alps and Pyrenees.

A detailed point-by-point response to every comment is provided below along with all the proposed changes in the revised manuscript. Line numbers refer to the first submitted version of the manuscript.

The most important issue I concern is that how did the author performed the BC and dust deposition during 1979-2018 by using regional climate model. Snow cover is different from glacier. Snow cover will melt and disappeared in summer season in these two mountain regions, which will not accumulate BC and dust deposition. From Figure S2, the author only showed the annual BC and dust deposition, I guess the author use the annual deposition of BC and dust and evaluate the impact on snow cover melting? How did the author input the BC and dust deposition date to the snow cover in the model?

In this study, we used the hourly deposition fluxes computed by the regional climate model CNRM-ALADIN63. These fluxes were distributed between wet and dry deposition according to the hourly precipitation values (dry deposition only if there is no precipitation, wet deposition if there is precipitation). Hourly meteorological variables are used as inputs in the snow model. These methodological developments are described in the method (lines 280 to 314). The hourly deposition fluxes are thus direct inputs to the snow model (Tuzet et al., 2017). This ensures an accurate distribution of the dust and BC particles inside the snowpack. This was mentioned in the method section (lines 287-292 of the original manuscript): "*BC and dust deposition fluxes are obtained from the regional climate model CNRM-ALADIN63 (Nabat et al., 2020). This model includes an interactive tropospheric aerosol scheme able to represent the main aerosol species such as BC and dust in the troposphere. These aerosols are prognostic variables, subject to transport, dry deposition and in-cloud and below-cloud scavenging. In this study, hourly output of dry and wet BC and dust deposition fluxes were used, coming from a simulation over the 1979–2018 period driven by the ERA-Interim reanalysis in order to ensure a realistic timeline of the evolution of aerosol deposition.*"

Our simulations are for seasonal snow only and not snow on glacier. This implies that all the LAPs accumulated in the snowpack are removed when the snow melt (see line 73 in the introduction). Following the reviewer comment, we have modified the beginning of the results section to make these two points more explicit (lines 66-69): "*Simulations explicitly accounting for dust and BC deposition fluxes are compared to a pure snow simulation, excluding any LAP. Simulations of the seasonal snow cover Simulations are driven by the hourly meteorological reanalysis S2M (Vernay et al., 2021) and hourly deposition fluxes of BC and dust from the regional climate model CNRM-ALADIN63 driven by reanalysis data, so as to follow the unfolding of observed meteorological conditions (Nabat et al., 2020).*"

We also added the information about seasonal snow cover in the methods (lines 347-371, section simulations description) : "*The model ran over the period 1979–2018, in a semi-distributed geometry (i.e., per 300m elevation bands, for each massif, following the SAFRAN geometry). For each simulation point, the snow cover fraction is set to 1 whenever snow is present. Note that only seasonal snow is simulated. In our simulations we are not accounting for snow on glacier. LAPs accumulated during one season are thus removed with runoff when the snow cover melts out.*"

The goal of Figure S2 is to compare the deposition of BC and dust from two models: CNRM-ALADIN63 (used in this study) and GFDL-AM4. Data from CNRM-ALADIN63 are available at hourly time step (as mentioned above), while GFDL-AM4 provides daily fluxes. For this Figure S2, data from both models were aggregated to the same annual time step in order to provide a comparison of the annual fluxes and inter-annual variability across these two models. The legend of Figure S2 has been modified as follows :

Annual BC and dust deposition. *Temporal evolution of the total annual BC (grey) and dust (orange) deposition from CNRM-ALADIN63 model (solid lines) and for the 1979-2018 period, at 2100 m a.s.l. Annual cumulative BC and dust are computed from hourly data from the CNRM-ALADIN62 model (solid lines) for the 1979–2018 period and from daily data from the GFDL-AM4 model (dots) for the 1980-2014 period. The shaded area from CNRM-ALADIN63 represents the standard deviation between all the grid points located at 2100 m a.s.l. for each the mountain region.*

For the current title, what kind of response of snow cover to black carbon and dust? I see from the results, the author wants to show BC and dust advanced the snow melting and runoff peak. The title should be revised.

The key finding of this paper is that BC and dust deposition on snow cover trends have modified the snow cover duration trends due to atmospheric warming alone. However, both reviewers pointed out that the explanations on this matter were confusing. We therefore provided a complete revision of the section "Past trends (1979-2018)". Please refer the details and changes provided below in response to referee 1 specific comments 11, 12 and 13 and referee 2 comment 2. We believe that the original title supports the key finding of the manuscript regarding the impact of BC and dust on snow cover trends. In addition, the main objective of the study has been clarified in the introduction, as requested in specific comment 1. Please see response to specific comments

For the abstract, no results on the hydrological impact by BC and dust, which is the most important part in this study.

We acknowledge that the hydrological impact is also an important conclusion of the manuscript. The abstract was modified to account for this comment and now reads:

"By darkening the snow surface, mineral dust and black carbon (BC) deposition enhances snowmelt and triggers

numerous feedbacks. Assessments of their long-term impact at the regional scale are still largely missing despite the environmental and socio-economic implications of snow cover changes. Here we show, using numerical simulations, that dust and BC deposition advanced snowmelt by 17 ± 6 days on average in the French Alps and the Pyrenees over the 1979–2018 period. *BC and dust also advanced by 10–15 days the peak melt water runoff, a substantial effect on the timing of water resources availability.* We also demonstrate that the decrease in BC deposition since the 1980s moderates the impact of current warming on snow cover decline. Hence, accounting for changes in light-absorbing particles deposition is required to improve the accuracy of snow cover reanalyses and climate projections, that are crucial for better understanding the past and future evolution of mountain social-ecological systems."

Specific comments

1.a. For the introduction, the brief review on research progress of BC and dust and their climate effect on snow melting was not completed (see below references). Especially in Arctic, North America, and High Mountain Asia, the research results and gaps were not mentioned and summarized, which limited this study to abroad audience from a global view of this topic.

1.b. Meanwhile, no description on the importance of the two regions- Alps and Pyrenees mountains, why you choose these two regions?

1.c. At the end of the introduction, no specific purpose of this study was delivered.

1.d. The reason why the author excluded the discussion on brown carbon and algae was not sufficient.

References. Kang S., Y. Zhang, Y. Qian, H. Wang. 2020. A review of black carbon in snow and ice and its impacts on cryospheric change. Earth-Science Reviews, 210, 103346. j.earscirev.2020.103346.

Kang S., Q. Zhang, Y. Qian, Z. Ji, C. Li, Z. Cong, Y. Zhang, J. Guo, W. Du, J. Huang, Q. You, A. K. Panday, M. Rupakheti, D. Chen, Örjan Gustafsson, M. H. Thiemens, D. Qin. 2019. Linking Atmospheric Pollution to Cryospheric Change in the Third Pole Region: Current Progresses and Future Prospects. National Science Review, 6(4): 796-809. <https://doi.org/10.1093/nsr/nwz031>.

Schmale, J., Flanner, M., Kang, S., et al., 2017. Modulation of snow reflectance and snowmelt from Central Asian glaciers by anthropogenic black carbon. Sci. Rep., 7:40501. DOI: 10.1038/srep40501 1

Zhang Y. L., S. Kang, M. Sprenger, Z. Cong, T. Gao, C. Li, S. Tao, X. Li, X. Zhong, M. Xu, W. Meng, B. Neupane, X. Qin, M. Sillanpää. 2018. Black carbon and mineral dust in snow cover on the Tibetan Plateau. The Cryosphere, 12: 413-431. <https://doi.org/10.5194/tc-12-413-2018>.

1.a. It is difficult to include all the above references since the maximum number of references allowed by the journal is 70. Despite the clear relevance of the studies by Kang et al., 2018, Schmale et al., 2017 and Zang et al., 2018, we do not have space to include them all. However, reference to Kang et al., (2020) was added to the manuscript as it includes the 3 other missing references and provides a review of the impact of BC on snow and ice. Note that reference to Kang et al., (2020) has been added at several other places in the manuscript.

Lines 45-46 in the introduction have been modified as follows : "*BC is the most efficient absorbing aerosol in the atmosphere (Flanner et al., 2017, Skiles et al., 2018) and its deposition strongly impacts snowmelt rates (Skiles et al., 2018), as shown for instance in High Mountain Asia, Europe, North and South America (Kang et al., 2020 and references therein). Yet the contribution of the regional changes in BC deposition to the observed snow cover decline remains to be assessed (Hock et al., 2019).*"

1.b. This study exemplifies the impact of light absorbing particles on seasonal snow cover in mid-latitudes regions, using the French Alps and Pyrenees as complementary case studies, two mountain ranges of high relevance for water resources and socio-economics. This is now detailed in subsection "Study sites: French Alps and Pyrenees" in the Methods section (line 272) and read as follows:

The French Alps correspond to the headwaters of the Rhône river, one of the major rivers in Europe, and many of its tributaries ; the Pyrenees correspond to the headwaters of several major rivers in southern France (Garonne and Adour rivers) and northern Spain (Ebro river). The French Alps and Pyrenees experience different climatic conditions, spanning the Mediterranean to continental climates in Europe. The French Alps and Pyrenees host the vast majority of the ski resorts in France. French ski industry is ranked third globally in terms of skier visits over the past decades. The changes in snow cover melt-out date spanning the French Alps and Pyrenees is thus of substantial relevance for the national ski industry. This case study thus provides a basis for further studies carried out at the European scale or addressing other mountain regions in the future. "

More synthetic information was added in the main text lines 57-64 :

"Regional assessments of the combined effect of LAPs deposition on snow cover changes over long time periods are currently missing (Skiles et al., 2018; Hock et al., 2019) and require an accurate representation of the snow-albedo feedback triggered by the presence of LAPs. The aim of this study is thus to quantify the combined effect of BC and dust deposition on snow cover dynamic and trends at the regional scale of the French Alps and Pyrenees mountain ranges, for the 1979–2018 period. The snow cover represents a major driver of hydrological processes in both mountain ranges. Its variability exerts a strong influence on water resource availability in southern France and northern Spain but also on the income generated by winter tourism in both countries. Considering the large uncertainty in brown carbon deposition and absorption efficiency (Skiles et al., 2018), the effect of brown carbon is excluded from the analysis as well as the effect of algae that generally bloom only at high elevation and late in spring (Di Mauro et al., 2020, see the Limitations section. Our analysis is based on numerical simulations performed with the detailed snow cover model Crocus accounting for the complex interactions between LAPs and snow (Tuzet et al., 2017). Hence, this study exemplifies the impact of dust and BC on seasonal snow cover in mid-latitudes mountain regions."

1.c. The aim of the study has been clarified in the revised version of the manuscript. Please see the proposed modifications in the response to comment 1.b.

1.d. The choice to exclude the brown carbon and algae in this study is briefly introduced in the introduction (lines 60-62) and was detailed in the section "limitations" of the manuscript (lines 219-226) as reported below: *In this study, we considered only two types of LAPs, BC and dust, while other types may also contribute to modify the radiative forcing on the snow cover. Brown carbon was excluded considering the high uncertainty in atmospheric concentrations and optical properties estimates (Skiles et al., 2018). Snow algae could play an important contribution to snow cover melt in some regions such as Greenland (Hoham et al., 2020) but their impact on seasonal snow in the European Alps is not yet bounded and their impact is limited to the very end of the snow season and for high elevations (typically above assumed to hold the dominant role in accelerating snow melt. BC and dust optical properties vary considerably (Tuzet et al., 2019; Caponi et al., 2017).*

To make this point clearer, considering the limitation in the number of words of the main text and the already quite long introduction, we chose, in the revised version, to refer to the section limitations in the introduction. Please see proposed changes in response to comment 1.b.

2. The snow cover area and snow depth simulations were evaluated using satellite data and in situ observations. BC and dust deposition flux data used for model simulation also should be evaluated or the uncertainties should be addressed. It seems like the dust deposition used in this study was overestimated compared with other study in Figure S2.

Due to the lack of observations of BC and dust deposition fluxes, especially at large scale, evaluating LAP deposition fluxes from models is still very challenging. The depositions fluxes used in our study, coming from CNRM-ALADIN63 model, have been evaluated at one location in the French Alps by Tuzet et al., (2020, indirectly through snow cover simulation results.). The authors compared the radiative impact of BC and dust between simulated with Crocus forced with CNRM-ALADIN63 deposition fluxes and in-situ spectral reflectances for two snow seasons (Figure 5a in Tuzet et al., 2020). They reported an R^2 of 0.78 between simulated and measured radiative impact and a mean bias of 31 ng g^{-1} in optically equivalent BC surface concentration (results slightly underestimate the observed values). The direct comparison between LAP chemical measurement in snow and their radiative impact is highly uncertain due to strong uncertainties both on LAP measurements in snow (e.g. Figure 3a and 4 in Tuzet et al., 2020) and on LAP mass absorption efficiencies (MAEs) from the literature. However in the present study, the key relevant variable is the radiative impact of BC and dust, since we aim to correctly simulate the snowmelt. The radiative impact of BC and dust was indirectly evaluated by comparing snowmelt rates and snow depth simulated and observed using almost 500 snow depth measurements stations (Fig. S8). The accurate simulation of snowmelt and snow melt out date (please see Response to reviewer 2, specific comment 1) provides confidence in the combined deposition fluxes and MAE used in the study.

We modified the section "Limitations" in the section methods (lines 432-441) as follows: *"Still, BC and dust deposition from CNRM-ALADIN63 have been evaluated by Tuzet et al., (2020) at Col du Lautaret (French Alps) over two snow seasons. Using in-situ spectral reflectances, they evaluated the radiative impact of BC and dust simulated with Crocus using CNRM-ALADIN63 model deposition fluxes as inputs. They reported an R^2 of 0.78 and mean bias in BC equivalent surface concentration of 31 ng g^{-1} results slightly underestimate the observed values). Such an evaluation at one location is not sufficient. In our study, we indirectly evaluated the simulated radiative impact of BC and dust by evaluating SMOD and snowmelt rates using 495 snow depth measurement stations (Table S1 and Figure S8). For both SMOD and snowmelt rates, the simulations using BC and dust from CNRM-ALADIN63 outperform the other simulations. This provides further confidence in the overall accuracy of the simulated radiative*

impact, i.e. the combination of the deposition fluxes and the optical properties of BC and dust (see Model uncertainties). Figure S_2 compares the annual BC and dust deposition values from CNRM-ALADIN63 and from the global model GFDL-AM4 (Zhao et al., 2018a,b). BC deposition and trends values from CNRM-ALADIN63 are in agreement with BC deposition modeled by GFDL-AM4 (mean bias of 0.004 g m^{-2} , corresponding to about 9% of the mean BC loads) while dust loads from CNRM-ALADIN63 are higher by a mean factor of 3.7 compared to the ones from GFDL-AM4. This might be related to the coarser spatial resolution of GFDL-AM4. Both models indicate no significant trend in dust deposition over the period 1980–2014."

The simulated radiative impact of BC and dust can suffer from the uncertainties in dust and BC deposition fluxes but also from the uncertainties on their optical properties. We therefore performed sensitivity tests to the optical properties of dust and BC. This is detailed in response to the specific comment 14.

3. In line 76, "Shortening of the season due to BC and dust" is not clear, which season?

The title of this section has been re-written accordingly and now reads :
"Shortening of the duration of the seasonal snow cover due to BC and dust"

4. Lines 82-84, the logic of this sentence is not right. From this sentence and Figure 1, No results indicated the variations of BC and dust reduced SMOD.

We agree that the sentence was confusing. We meant that SMOD reduction due to BC and dust varies from one year to the other, since dust and BC deposition varies from one year to the other. The sentence was re-written as follows : *"The SMOD reduction due to BC and dust deposition (i.e. delta SMOD on Fig. 1) varies from 0 to 47 days. The magnitude of the SMOD reduction shows strong inter-annual variations (Supplementary Fig. S1), due to the strong interannual variation of BC and dust deposition rates (Supplementary Fig. S2). The SMOD reduction also depends on the location and the elevation (Supplementary Fig. S11, S12)."*

5. Lines 93-95, the sentence is not completed. Also simulated results show a larger absolute impact of BC and dust on the SMOD at higher elevations, and authors explain this is associated with later melt and longer snow season at higher elevations, where dust and BC amplify its effect on snowmelt due to exposed to higher incoming solar radiation. Is this mechanism can be revealed in this snowpack model or only speculation?

Thanks for spotting the problem in lines 93-35. The sentences have been rewritten in response to comment 14 (please see proposed changes in the response to comment 14).

The mechanism behind the larger absolute effect of LAPs at high elevation was demonstrated by Dumont et al., 2020 using an ensemble version of the Crocus model. They compared the shortening of the snow cover duration simulated at several elevations for a given dust mass deposition and demonstrated that higher elevation are systematically more affected (Figure 7 in Dumont et al., 2020). Then, they compared the different components of the surface energy balance (net shortwave, longwave and turbulent fluxes) at different elevations simulated with Crocus with and without dust (Figure 8 in Dumont et al., 2020). They showed that the net shortwave input to the surface energy balance is higher at high elevations, while all the other fluxes are equal or lower. Lines 85-87 were thus modified as follows :

"This is related to later melt and longer snow season, where dust and BC are exposed to higher incoming solar radiation, amplifying its effect on snowmelt acceleration. This mechanism was deciphered using snowpack simulations by Dumont et al., 2020."

6. Figure 2 presented the parameters contribution to SMOD (snow melt-out date), including weather, BC, and dust. Here, using "climate" is more appropriate than "weather".

We understand that the term "weather", used here to refer to the atmospheric conditions (temperature, precipitation, wind-speed, relative humidity) excepted light absorbing depositions, was ambiguous, and we replaced it with "meteorological conditions" consistent with the framing and terminology used in the body of the manuscript. Figures 2 and S4 were modified accordingly.

7. In the second part of the results (Figure 2), the contribution of meteorological condition to SMOD was much larger than BC and dust. The importance of BC and dust on SMOD in these two mountain regions need be clarified further.

We agree with the reviewer that the meteorological conditions are the primary driver of SMOD. In Figure 2, we investigate the relative contribution of meteorological conditions and LAPs on the SMOD interannual variability (see line 105 to 129). We show that LAPs account for up to 30% of the variance of the SMOD, the largest part of the SMOD interannual variability being explained by meteorological conditions (e.g. small winter snowfall causes early SMOD). The importance of BC and dust on SMOD is discussed in the part 1 (lines 76 to 104, section "**Shortening of the duration of the seasonal snow cover due to BC and dust**", while part 2 of the results (lines 105 to 129, section "**BC and dust effect on snow cover inter-annual variability**" and figure 2 refers to the effect on the interannual variability.

8. The impacts of BC and dust on snow cover changes by analyzing SMOD changes were quantified only. Other parameters, such as snow surface albedo changes, can be a directly evidence for quantifying the impacts of light-absorbing particles on snow cover and should be discussed.

We initially chose to present the results in terms of SMOD changes only since we believed it is a synthetic variable, easy to understand and to use by scientists outside the snow community (e.g. ecologists). However, we fully acknowledge that additional variables such as albedo change or radiative impacts might also be useful. As a consequence, both changes in albedo and in absorbed solar energy were calculated and two figures were added in the supplementary (Fig S9 on albedo and Fig S10 on radiative impact, see below).

The main results on albedo change and addition energy absorbed due to BC and dust are briefly summed up in the main text (line 82) : "These averages are calculated accounting for the relative surface at each elevation band in both mountain ranges. At local noon, BC and dust decrease the mean surface albedo by 0.03 to 0.05 on average depending on the area (Supplementary Fig S9) and increase the solar energy absorbed by the snowpack by 7 to 13 \$\text{W m}^{-2}\$ on average (Supplementary Fig S10). The SMOD reduction ... "

Figure 1: **Figure S9 : Broadband albedo changes.** Differences between pure and impure simulations (i.e. BC + dust, only BC and only dust) of the mean albedo at 12h - local time. The differences are computed at 2100 m a.s.l., considering the entire study period (i.e., 1978-2018). Daily differences are computed over the snow season (i.e. for days with snow on the ground) and the average of the annual difference is used for the figure. The boxes show the quartiles of the distribution corresponding to the inter-annual and spatial variability. Minimum/maximum ranges (excluding outliers) are indicated by the whiskers.

Figure 2: Figure S10 : **Radiative impacts**. Differences between pure and impure simulations (i.e. BC + dust, only BC and only dust) of the mean solar energy absorbed at 12h. The differences are computed at 2100 m a.s.l., considering the entire study period (i.e., 1978-2018). Daily differences are computed over the snow season (i.e for days with snow on the ground) and the average of the annual difference is used for the figure. The boxes show the quartiles of the distribution corresponding to the inter-annual and spatial variability. Minimum/maximum ranges (excluding outliers) are indicated by the whiskers.

9. The statements in lines 111-119 were not clear and should be strengthened. It seems like that aged snow related to meteorological conditions can lead to the enrichment of light-absorbing particles and had stronger impacts on snow cover, and the contribution of this effect was separated and analyzed in this part? And how to separate the contribution of BC and dust only without considering this effect?

What we wanted to show in lines 111-119 is that the sequencing between meteorological conditions and deposition of dust and BC exerts a strong influence on the ultimate impact on snow cover. Surface enrichment in LAPs when meteorological conditions are favourable to melt is an example of positive feedback caused by the interactions between LAPs deposition and meteorological conditions. Another example described in this paragraph is the dependence of LAP impact to subsequent meteorological conditions. Whether LAPs remains near the surface or are buried by new snow strongly influence its radiative impact. Tuzet et al., 2020 also compared two snow seasons at the same location with contrasted LAPs deposition and showed that in both cases the shortening of the snow season attributed to LAPs is almost the same due to differences in meteorological conditions between the two seasons. Tuzet et al., 2020 also demonstrated that for these two snow seasons, the indirect radiative effect of LAPs, i.e. accelerated snow metamorphism, is low compared to previous studies (Tuzet et al., 2017 ; Skiles and Painter, 2019) and this again is due to meteorological conditions. Using the ANOVA analysis method (e.g. Geoffroy et al., 2012), we separated different process contribution to the SMOD variance: LAP deposition only, meteorological conditions only and interaction between the two.

Following Eq. 6 in Geoffroy et al., 2012, the SMOD, S , can be decomposed as a sum of one-variable functions and an interaction term I :

$$S = p_0 + p_1(\text{meteo}) + p_2(\text{LAP}) + I(\text{meteo}, \text{LAP}) \quad (1)$$

where p_0 is constant, p_1 depends on the meteorological conditions only, p_2 depends on LAPs deposition fluxes only and I is an interaction term called sequencing, including first and higher order interactions between the meteorological conditions and LAPs deposition (amount and timing). Please see supplementary material for the details on how we compute the different terms. Lines 111-119 discuss the contribution of the different terms. This paragraph was fully rewritten and the supplementary text on ANOVA was extended. Modifications in the supplementary and main text are listed below in blue.

"A. ANOVA equations

Following Eq. 6 in Geoffroy et al., 2012, the SMOD, noted S , can be decomposed as a sum of one-variable functions and an interaction term I , referred to as a "sequencing" term:

$$S = p_0 + p_1(\text{meteo}) + p_2(\text{LAP}) + I(\text{meteo}, \text{LAP}) \quad (2)$$

where p_0 is constant, p_1 depends on the meteorological conditions only, p_2 depends on LAPs deposition fluxes only and I includes interactions between meteorological conditions and LAPs deposition, i.e., the fact that the response to given LAP deposition can depend on meteorological conditions. For example, dust might be directly buried after deposition if a sufficiently large snowfall event occurs immediately after dust deposition.

The contribution, \hat{c} , of each function p_1 , p_2 to the variance of the SMOD is computed following the equations (2) and (3), where N_1 denotes the number of values taken by the meteorological conditions (39 individual years) and $N_2 = 2$ (simulations with or without LAPs). i, j are the parameter values. The interaction term (\hat{c}_I) (or sequencing) is computed following the equation (4).

$$\hat{c}(p_1) = \frac{1}{\text{var}(S)} \frac{1}{N_1} \sum_{i=1, N_1} \left[\frac{1}{N_2} \sum_{j=1, N_2} (S_{i,j} - \bar{S}) \right]^2 \quad (3)$$

$$\hat{c}(p_2) = \frac{1}{\text{var}(S)} \frac{1}{N_2} \sum_{i=1, N_2} \left[\frac{1}{N_1} \sum_{j=1, N_1} (S_{i,j} - \bar{S}) \right]^2 \quad (4)$$

$$\hat{c}(I) = 1 - (\hat{c}(p_1) + \hat{c}(p_2)) \quad (5)$$

"

Lines 111-119 was modified as follows: "For given deposition fluxes of BC and dust, the impact on the snow cover depends on the meteorological conditions (Tuzet et al., 2020). For instance, the same amount of BC and dust

deposited at the snow surface has a stronger impact if a long time period without snowfall follows the deposition, than when a new snowfall buries the particles immediately after deposition. Using the ANOVA method, we can separate two types of contribution to the SMOD variance, the contribution of LAPs deposition observed for average weather conditions, and the contributions through interactions between meteorological conditions and LAP deposition. The later is named the sequencing part (see Supplementary text). This sequencing of meteorological conditions and BC and dust deposition contributes to 2.9 – 9.1% to the total variance, while the deposition only contributes to 12.7 – 25.5% of the total variance (Fig. 2). Hence, deposition of dust and BC markedly contributes to the variance of the SMOD and this is partly due to the sequencing between the meteorological conditions and the deposition."

10. Line 128, What is "atmospheric trends"?

We agree that "atmospheric trends" is confusing. Here we referred to trends over the last four decades in all the atmospheric variables used as inputs: air temperature, BC and dust deposition fluxes, etc ... The sentence was changed as follows:

"The calculation of their contribution has been performed on the 39 years time series without detrending and thus includes the effects of trends over the last four decades in the input atmospheric variables, such as air temperature, BC and dust deposition fluxes."

11. Discussion on the "past trends", in the result of Shortening the season due to BC and dust, the Figure 1 showed the impact of BC and dust on SMOD was larger in the high elevations (>3000m), then why discussed the results at 2100 m asl?

The elevation range around 2100m a.s.l. discussed in this section represents about 20% of the French Alps and 40% of the Pyrenees area. Higher elevations represent a very small fraction of the Alps and Pyrenees (i.e. >2700m a.s.l. is about 6% in the Alps and 3% in the Pyrenees) and are generally concentrated over a reduced area. This means that the results in the 2100 m elevation band have stronger implications on water resource availability.

As this elevation was also chosen for the ANOVA study, we added a justification of the choice in the revised manuscript, in the previous section "BC and dust effect on snow cover inter-annual variability" (line 110):

"We discuss the results at 2100 m a.s.l, since 20% of the surface of the French Alps and 40% for the Pyrenees lies in this elevation band, making it particularly relevant for water resources and ecosystem disturbances."

Finally, the past trends analysis is performed at all elevation (Supplementary text and Fig. S3) and the conclusions drawn at 2100 m a.s.l. also stand at other elevations. This was mentioned in the text lines 150-152.

12. Lines 145-146, the sentence is difficult to understand. When BC and dust deposition to the pure snow, BC will enhance the melting rate. If less BC deposition, the melting rate became smaller, but still enhance the melting rate compared to pure snow, rather than "decrease melting rate". You can check the reference Flanner et al., 2007; Skiles et al., 2018.

We agree that the sentence lines 145-146 could be clarified. The sentence was thus changed as follows: *"In this recent period, the deposition of BC is reduced compared to the 1980s. BC deposition accelerates melt less than in the 1980s. Thus, the reduction of BC deposition fluxes partly offsets the effect of rising temperature in the 2009-2018 period."*

The entire section **Past trends** was rewritten, please see comment 2 of referee 2 for details and proposed changes.

13. Lines 147-148, the dust did not significantly affect the SMOD trend. Why the dust affected the SMOD, then had almost no effect on SMOD trend? Please explain more on this sentence. The Figure 3 and Figure S2 is kind of paradox. Make them clearer.

We agree that our explanation was confusing. We meant that dust deposition only slightly affect the SMOD trend (compared to BC that affects largely the trend because of a significant decreasing trend in BC deposition). In response to this comment and to comment 2 of reviewer 2, we entirely revised the section **Past trends** and the changes are reported below.

"Past trends (1979–2018)

The simulated SMOD for snow with BC and dust shows a negative trend over the last decades (1979–2018)

indicating earlier snowmelt (Fig. 3). In this section, we mostly discuss the results in the North Alps since this is the only area where most of the trends are significant. In the Northern Alps, the SMOD at 2100 m a.s.l. decreases by 5.7 days per decade (corresponding to a decrease of 2.9% of the snow season duration), which is consistent with Matiu et al., 2021. This decrease is consistent with the atmospheric warming rate. This warming rate ranges between +0.25 and 0.3°C per decade according to the reanalysis data used in this study (Fig. 3).

The simulated SMOD trend for pure snow is more pronounced than for snow with BC and dust. At 2100 m a.s.l., the pure snow SMOD decrease is 7.8 days per decade for the North Alps. The deposition of dust and BC thus alters the response of snow to warming. First, the sensitivity of SMOD to meteorological conditions is not the same for pure snow and for snow containing LAPs (e.g. Fig. 8 in Dumont et al., 2020). Second, the dust and BC inputs were not constant over the last four decades. BC deposition exhibits a negative trend, with a decrease of 11% on average considering all the massifs. Simulations accounting for BC deposition yield a negative SMOD trend lower than the SMOD trend of the pure snow simulation (Fig. 3). This means that, at the beginning of the study period (1979–1988), the shortening attributed to BC deposition alone (i.e. 15 days on average for all the areas) is higher than for the recent period (2009–2018) (i.e. 10 days only in average). In this recent period, the deposition of BC is reduced compared to the 1980s. BC deposition accelerates melt less than in the 1980s. Thus, the reduction of BC deposition fluxes partly offsets the effect of rising temperature in the 2009–2018 period. Conversely, no significant trend in dust deposition is identified. Dust deposition consequently affects the SMOD trend less than BC deposition does (Fig. 3).

Hence, the deposition of BC and dust modifies the SMOD trend in response to climate warming. In the North Alps, the primary driver of the trend modification is the decrease in BC deposition since the 1980s. This decrease compensates part of the impact of warming on the trend of the snow cover duration at 2100 m a.s.l. and is also observed for all the Alps and Pyrenees to a lesser extent (Supplementary text and Fig. S3). However, the trends presented in this section originate from a single model and the limitations of this study are discussed below."

14. It was concluded that the effect of BC deposition on SMOD was prevailing over the dust effect. But the uncertainties due to mass absorption efficiency, deposition flux, snow cover fraction, scavenging efficiency in snow were discussed relatively less. Those are fundamental settings to models of radiative transfer and could cause large biases.

We agree with the reviewer that these are fundamental settings of the model. The model uncertainties are discussed in the Methods Section in the section entitled Model uncertainties (line 463 to 511). A full section is dedicated to the uncertainties of mass absorption efficiency (line 464–485) and another section (line 486–493) is dedicated to the scavenging efficiency effect (please see also the additional sensitivity figure provided below). Regarding the deposition fluxes, their uncertainties is assessed together with the uncertainties of the selected optical properties in response to comment 2 and to comment 1 of referee#2. Please see response to both comments for proposed changes in the manuscript. Finally in the model configuration due to the geometry of the simulations (see response to comment 2 and discussion concerning the associated uncertainties lines 505–511), no snow cover fraction was used. This information was added line 322 which now read:

"The model ran over the period 1979–2018, in a semi-distributed geometry (i.e., per 300 m elevation bands, for each massif, following the SAFRAN geometry). For each simulation point, the snow cover fraction is set to 1 whenever snow is present on the ground."

The section on the uncertainties on the mass absorption efficiency and on the scavenging coefficient were partly rewritten to be clearer as follows:

1. Optical properties of BC and dust

The choice of dust refractive index and mass absorption coefficients also implies some uncertainties. In this study, values for Saharan Libya PM2-5 (Caponi et al., 2017) are selected: a mass absorption efficiency at 400 nm of $110 \cdot 10^{-3} \text{ m}^2 \text{ g}^{-1}$ and a dust Angstrom exponent equal to 4.1. This choice is made as Saharan dust is the primary source of dust deposition in European mountainous areas and as Tuzet et al., 2020 demonstrated based on spectral reflectance measurements that this spectral signature agrees well with the measured spectrum for two winter seasons. However, this spectral signature is expected to vary with the source location which also varies over time. Simulations performed by changing this spectral signature, with a mass absorption efficiency at 400 nm ranging between $27 \cdot 10^{-3} \text{ m}^2 \text{ g}^{-1}$ (corresponding to the source Sahel – Bodélé PM10) and $630 \cdot 10^{-3} \text{ m}^2 \text{ g}^{-1}$ (for dust PM2-5 from Sahel – Mali). The Angstrom exponent is equal to 3.3 (Bodélé) and 3.4 (Mali). These two indices are selected as they represent low and high values absorption for African dust (Caponi et al., 2017). This sensitivity study allows quantifying the uncertainty related to the spectral signature chosen (Supplementary Fig. S5). Depending on this parameterization, the median of the ΔSMOD

at 2100 m a.s.l. varies between 15 to 25 and 12 to 23 for the North and South Alps respectively, and between 11 to 20 and 10 to 21 for the North and South Pyrenees respectively. The median of the Δ SMOD with the spectral signature chosen in this study is closer to the lower median (i.e. 19 (15) for the North (South) Alps and 13 for both the North and South Pyrenees). This suggests that the effect of dust reported in this study might be under-estimated for Saharan dust deposition events originating from locations close to Mali. Finally, uncertainties are also associated with the modelling of BC absorption efficiency in snow. Indeed, the evolution of this variable is still poorly understood with variations of at least a factor 2 reported in the literature (e.g. Tuzet et al., 2019 and references therein). We thus estimate that the uncertainties associated with the selected optical properties of LAPs range within ± 5 days for Δ SMOD (Fig. S5).

2. Scavenging of BC and dust

In this study the scavenging of impurities in the snowpack was disabled as in Tuzet et al., 2020. Indeed, the lack of quantitative observations of LAPs percolation at the snow surface in presence of melt water does not allow a proper evaluation of this effect. A sensitivity test to the scavenging coefficient was performed allowing 20% of the BC (5% of the dust) to be scavenged with the water percolation (Tuzet et al., 2017). This leads to a small effect on the value of Δ SMOD (i.e. a median of the Δ SMOD variation at 2100 m a.s.l. of 2 days for the North Alps and 1 day for the South Alps).

The Δ SMOD differences between the BC only and the dust only simulations range between 2 and 5.5 days for the North Alps and are smaller for the North Pyrenees. The paragraph on how BC prevails on the North of the mountain ranges (lines 88-95) was thus revised as follows :

"The effect of BC is likely prevailing in the Northern part of the French Alps, where higher BC deposition is observed (Supplementary Fig. S2). The differences in Δ SMOD between the BC only and dust only simulations are similar to the uncertainties caused by the uncertainties on dust optical properties of dust (see Method Section). The simulated SMOD when accounting for BC is earlier on average by 14.2 days (7.1%) in the North Alps and by 13.3 days (6.9%) in the South Alps, compared to the pure simulation. The southern parts of both mountain ranges are more exposed to southerly winds carrying Saharan dust (Supplementary Fig. S2), hence exhibit larger dust depositions than their northern counterparts."

The concluding remarks were also modified as follows :

"Our results demonstrate that BC and dust deposition advances the end of the snow season by 17 ± 6 days on average over the French Alps and the Pyrenees mountain ranges and the peak snowmelt runoff by two weeks. Given the relevance of the snow cover in terms of water resources and tourism in these regions, both effects have deep socio-economic implications. It is also a significant effect to consider in mountain ecosystems studies. The BC effect is likely prevailing over the dust effect in the North Alps. BC deposition alone shortens the snow season by 11 days on average, compared to pure snow simulations. However, the BC effect would be even larger without the decrease in BC deposition observed since the 1980s."

Finally in the main text, line 219 to 239 discussing of the uncertainties of the simulation were rewritten as follows :

"In this study, we consider only two types of LAPs, BC and dust, while other types may also contribute to modify the radiative forcing on the snow cover. Brown carbon is excluded considering the high uncertainty in atmospheric concentrations and optical properties estimates (Skiles et al., 2018). Snow algae play an important role in the snow cover energy balance in Greenland (Hoham et al., 2020) but their impact on seasonal snow in the European Alps is not yet bounded and remains limited to the very end of the snow season at high elevations (typically above 2000 m a.s.l.) (Di Mauro et al., 2020 ; Stewart et al., 2021). BC and dust are thus assumed to hold the dominant role in accelerating snow melt in the studied mountains. BC and dust optical properties are reported to be strongly variable (Bond et al., 2006, Tuzet et al., 2019 ; Caponi et al., 2017). The absorption efficiencies values selected here are intermediate values between extremes found in the literature and were evaluated at one location in the French Alps (Tuzet et al., 2020). The sensitivity to dust refractive index is investigated using very high and low values for absorption from Caponi et al., 2017 (see Supplementary Fig. S5 and Methods) resulting on a maximum uncertainty of ± 5 days for Δ SMOD. The additional uncertainty associated to the choice of a scavenging coefficient for BC and dust is estimated to 1 to 2 days (see Methods)."

"The uncertainties on the results also originate from the choice of a given snow model (Lafaysse et al., 2017). This is quantified using an ensemble approach with 35 configurations of the Crocus snow model. The deterministic simulation gives a slightly lower estimate of the impact of BC and dust on SMOD compared to the ensemble simulation (Supplementary Fig. S6). However, both estimates are close and the temporal trend is conserved. Finally, the impact of BC and dust deposition on snow is estimated from offline simulations and thus does not include neither the aerosol-atmosphere interactions nor the coupling between the surface and the atmosphere. Aerosols might cool

the atmosphere by scattering the solar radiation before being deposited onto the surface, while snow cover reduction can lead to positive feedback enhancing the atmospheric warming by advection of turbulent heat (Liston, 1995, see Method Limitations). Further research is required to investigate the simultaneous responses of the snow cover to these processes especially the potentially compensating effect of the particles in the atmosphere. In this study, we thus estimated that the interval of confidence associated with the average SMOD changes attributed to BC and dust is ± 6 days."

Please see also changes in response to comments 16 and 18 for the assessment of the uncertainties and the related changes in the manuscript.

Figure 3: Shortening of the season computed as the difference in snow melt-out date (SMOD) between the pure snow simulations and simulations considering BC and dust, considering the scavenging of BC and dust and without considering the scavenging (this study). SMOD differences are computed at 2100 m a.s.l., considering the entire study period (i.e., 1978-2018), for the North and South Alps. The boxes show the quartiles of the distribution corresponding to the inter-annual and spatial variability. Minimum/maximum ranges (excluding outliers) are indicated by the whiskers.

15. The SURFEX/ISBA-Crocus snowpack model is driven by the meteorological reanalysis S2M, whereas the BC and dust deposition fluxes from the CNRM-ALADIN63 model is driven by the ERA-Interim reanalysis. How do you evaluate the impact of uncertainties in using different meteorological reanalysis datasets to quantify respectively the deposition fluxes of BC and dust and its effects on snow cover change?

We agree that the use of two different reanalyses is prone to uncertainties. However, the same approach was evaluated in detail using field measurements in Tuzet et al., 2020. In addition to what was done in Tuzet et al., 2020, we used a downscaling method to obtain LAPs deposition from CNRM-ALADIN63 in S2M geometry (for more details of S2M geometry, please refer to comment 16). The limitations of combining both reanalyses and of the downscaling method were discussed and quantified in detail in the Method section (lines 442-462). This section was fully rewritten to be clearer.

The downscaling method for dust and BC deposition fluxes is also a source of uncertainties. We tested the impact of the number of points selected in each massif to compute the gradient of deposition fluxes. Results shows that differences in total BC and dust deposition are lower than 5%. The methodology to redistribute dry and wet deposition according to SAFRAN precipitation might also lead to uncertainties. We performed of sensitivity test with two extreme scenarios (i) all the fluxes are considered as dry deposition, (ii) all the fluxes are considered as wet deposition. Scenario (i) leads to a mean difference of 1.8 (± 1.4) days with the simulations used in the study. For scenario (ii), the mean difference is - 10 (± 2.3) days. These scenarios provide the uppermost and lowermost boundaries of the uncertainties due to this dry/wet partitioning. Even in such extreme cases, the difference is lower than the 17 days of shortening attributed to dust and BC.

Please see also responses to comments 14, 16 and 18 for discussion on the uncertainties and proposed changes.

16. The meteorological reanalysis S2M (one data point every 300 m) is used to drive the snowpack model. What is the spatial resolution of the SURFEX/ISBA-Crocus simulation here? How do you avoid the biases due to large resolution differences between different datasets used in the snowpack model, i.e., S2M (300m), BC and dust deposition flexes from the regional climate model CNRM-ALADIN63 (12km)?

The meteorological reanalysis S2M is not a 300 m grid. As described in Vernay et al., (2021) S2M operates "over elementary areas specifically designed to represent the main drivers of the spatial variability in mountain ranges called "massifs". A "massif" is a conceptual, object corresponding to a mountainous area (of about 1000 km² on average) over which the meteorological conditions are considered homogeneous for a given elevation. This hypothesis simplifies the representation of a complex topography by covering the different elevations of a given massif with a minimum number of representative computation points. The S2M reanalysis uses a 300 m vertical resolution. The upper (resp. lower) elevation for each massif is defined as the 300 m multiple immediately above the highest point (resp. below the lowest point) of a 50 m digital elevation model from the French National Geographic Institute (IGN) in the considered massif. In our study, one simulation is performed for each elevation band in each massif.

This geometry is equivalent to a semi-distributed representation often used in hydrological models. We added more information in the method section (lines 282-285) to clarify this point.

These data are assumed to be homogeneous within a given "massif". A massif is a conceptual object corresponding to a mountainous area (of about 1000 km² on average) over which the meteorological conditions are considered homogeneous at a given elevation. In this geometry, the Alps and the Pyrenees are both divided into 23 massifs (Fig S7) selected for their climatological homogeneity (Durand et al., 2009). The S2M reanalysis uses a 300 m vertical resolution (Vernay et al., 2021). In our study, for each massif, meteorological data depends only on elevation (one data point every 300 m) considering a flat surface.

As the LAPs deposition was available at 12 km grid resolution, a downscaling method was applied to obtain BC and dust deposition in the same geometry as S2M. This downscaling method is described in the method section. Note that different approaches have been tested for this downscaling and the uncertainties associated with this method have been evaluated, as mentioned in response to specific comment 15.

17. Authors only show the simulated results on average in the North and South Pyrenees (left) and the North and South Alps, respectively. It is better to display the results on the grid points, helping readers to understand the spatial heterogeneity of the effects of BC and dust on snow cover change and its hydrological implications.

As mentioned in the response to the comment 16, the spatial resolution of the simulations is not a grid, but

massifs with elevation bands. Such spatial representations (North, South Alps, ...) also enables to reduce the uncertainties. Considering the different sources of uncertainties discussed in the method limitations, we believe that discussing the results for these four large areas makes more sense than trying to read into the differences between the individual SAFRAN massifs. However, we acknowledge that the results per massifs could also be interesting. We consequently added the two supplementary figures below displaying the results of Figure 1 per massif. Note that the meaning of massif numbers is provided in Fig. S7. Supplementary figures S11 et 12 are quoted in the main text line 94.

Figure 4: Fig S11: Shortening of the season computed as the difference in snow melt-out date (SMOD) between the pure snow simulations and simulations considering BC and dust (red), dust only (orange), BC only (grey) and considering implicitly the LAPs (i.e. an albedo decrease based on the snow age only, baseline green). SMOD differences are computed at 2100 m a.s.l., considering the entire study period (i.e., 1978-2018), for each individual SAFRAN massif in the French Alps- see Fig. S7. The boxes show the quartiles of the distribution corresponding to the inter-annual variability. Minimum/maximum ranges (excluding outliers) are indicated by the whiskers.

Figure 5: Fig S12 :Shortening of the season computed as the difference in snow melt-out date (SMOD) between the pure snow simulations and simulations considering BC and dust (red), dust only (orange), BC only (grey) and considering implicitly the LAPs (i.e. an albedo decrease based on the snow age only, baseline green). SMOD differences are computed at 2100 m a.s.l., considering the entire study period (i.e., 1978-2018), for each individual SAFRAN massif in the Pyrenees - see Fig. S7. The boxes show the quartiles of the distribution corresponding to the inter-annual variability. Minimum/maximum ranges (excluding outliers) are indicated by the whiskers.

18. The authors discuss in detail the uncertainties of the driving datasets as well as parameters in the snowpack model. So many uncertainties may result in big simulation biases. Can you quantitatively assess these uncertainties? For example, previous study reported the S2M reanalysis underestimated of precipitation amount in the Alps. Meanwhile, the SMOD decreases are less than 3% of the snow season duration due to BC deposition in Line 142. If the decreased SMOD due the underestimated precipitation is greater than that from BC deposition, these simulation evaluations will be not enough to convince the readers. Thus, I suggest authors firstly quantified the uncertainties at the observation station. If the biases caused by the drivers and model parameters is acceptable, you then start to simulate in regional scale. It is better for authors to provide confidence interval in quantifying the effect of BC and dust on snow cover, to show the uncertainties of the simulations.

We thank the reviewer for this thoughtful suggestion. The meteorological reanalysis S2M uncertainties are discussed in detail in a recent paper which was not available at the time of the first submission (Vernay et al., 2021, Figs. 9 to 16). The simulated temperature and precipitation were evaluated with respect to in situ observations. The underestimation in precipitation noted by the reviewer is valid at very high elevation where no precipitation observation are assimilated in SAFRAN (Vionnet et al., 2019). The simulated snow depth (baseline simulations in our study) is also extensively evaluated in Vernay et al., 2021. The section on S2M uncertainties has thus been revised as follows (lines 421-430):

SAFRAN reanalysis

In snow modeling, most of uncertainties are brought by the meteorological forcing (Raleigh et al., 2015). Uncertainties in the SAFRAN meteorological reanalysis were assessed in details in Vernay et al. (2021). The uncertainties are variable in time and space. Queno et al. (2017) reported a bias of the shortwave radiation at some locations in the Pyrenees. Vionnet et al. (2019) showed an under-estimation of precipitation amount in the Alps at the highest elevations. Vernay et al. (2021) show that the temperature accuracy varies between 1 and 1.5 K and the accuracy of the monthly cumulated precipitation is roughly 20 kg m^{-2} with a low bias of less than 2 kg m^{-2} at 43 stations. However, in our study, the impact of meteorological input uncertainty is minimized as our conclusions are based on relative differences between simulations with identical meteorological forcing except LAP deposition and not on absolute values.

In addition, as suggested in this comment and in comment 1 of referee #2, we added to the manuscript an evaluation of the simulations (pure, BC+dust and baseline) using 495 snow depth measurement stations. The simulated and observed SMOD are now compared in Table S1 (see response to comment 1 of referee 2) showing an average bias à -1.7 day and a RMSE of 25.7 days (compared to 13.3 days and RMSE of 31.1 for the pure snow simulations). The melt rate were also evaluated at each station in the first version of the manuscript (Fig. S8). The very low bias of the BC+dust simulation at the 495 stations provides further confidence in our results. Note that the 3% mentioned in the comment (line 142) is a trend and note an absolute SMOD change (days per decade) and that the section **Past trends** have been rewritten in response to comment 13.

We also agree with the suggestion of providing a confidence interval for the SMOD change. Considering our response to comments 14, 15, 18 and comment 1 and referee 2, we estimate the maximum uncertainty associated with the average of 17 days shortening of ± 6 days (see Fig S5, Figure in response to comment 14).

The information was added as several places in the manuscript (in the abstract, see response to general comments, line 73-74 - see response to comment 1 of referee 2, in the concluding remarks, and limitations - see response to comment 14).

19. As for impacts on mountain hydrology, no significant change in the intensity of the peak runoff is attributed to the presence of BC and dust in Fig. 4. Can you explain the reasons accounting for this phenomenon, considering BC and dust lead to a shift in the CMD with earlier snowmelt?

This is a very interesting observation and we thank the reviewer for pointing it to us.

Whereas there is a clear shift in the peak melt water runoff timing, the magnitude of the peak runoff remains similar when LAPs are introduced in the model. By increasing the absorbed solar radiation, LAPs cause snowmelt to occur earlier in the snow season, when the available energy is lower (lower shortwave and longwave radiation, lower sensible heat flux). We hypothesize that this is the cause for the unchanged peak runoff magnitude despite the presence of LAPs on the snow surface. A similar mechanism causes a reduction of snowmelt rates under atmospheric warming (Musselman et al., 2017).

This was added in the section on "Impacts on mountain hydrology" as follows:

"BC and dust lead to a shift in the simulated CMD by up to 15 days earlier in the season (Fig. 4) over the 1979-2018 period compared to pure snow simulations. The effect is larger in the Alps (CMD shift of 15 days for the

Alps) than for the Pyrenees (CMD shift of 10 days) in agreement with the larger effect on snow cover duration in the Alps mentioned above. No significant change in the intensity of the peak runoff can be attributed to the presence of BC and dust (Fig. 4). By increasing the absorbed solar radiation, LAPs cause snowmelt to occur earlier in the snow season, when the available energy is lower (lower shortwave and longwave radiation, lower sensible heat flux). We hypothesize that this is the cause for the unchanged peak runoff magnitude despite the presence of LAPs on the snow surface. A similar mechanism causes a reduction of snowmelt rates under atmospheric warming (Musselman et al., 2017). Even if the magnitude of the peak runoff is preserved, earlier snowmelt can have a profound consequences on the management of the water resource and downstream alpine ecosystems (Bard et al., 2015)."

Reviewer #2

Main comment

Assessments of long-term impact of LAPs at the regional scale are few, and this manuscript aims to address the same over French Alps and the Pyrenees over the 1979–2018 period using offline numerical simulations. Their analysis illustrates that dust and BC deposition has advanced the region's snowmelt by 17 days on average which can be a major implications for water availability. The relative influence of dust and BC in elevational gradient and spatial variations of reduction in SMOD is aptly reported. Further, they analyse the interannual variability and attempts to separate out the BC and meteorology impacts. This is a well-organized and interesting study which is advancement in the field. However, a few things are not clear.

The authors are grateful for the very positive and encouraging feedback, which have been very helpful in improving the manuscript. All the comments were accounted for. We added a direct evaluation of the simulated SMOD using 495 measurement stations. We discussed the uncertainties due to the use of offline simulations. We revised the discussion on trends and we provide more details on the ANOVA analysis. The detailed answers to every specific comment and the proposed changes in the manuscript are listed below.

Specific comments

1. The entire analysis is based on simulated SMOD, hence detailed evaluation of simulated SMOD should be included.

We agree with the reviewer that a detailed evaluation on SMOD was missing. In the revised version, we provide this evaluation by comparing the simulated SMOD to the observed SMOD at 495 stations measuring snow depth in the French Alps and Pyrenees over the 1983-2018 period (Table S1, see below). Since SMOD is not available for every year at each station, the number of observations used was also added in the table. The results show that considering the explicit representation of BC and dust outperform the other simulation set-up with an overall bias of 1.7 days. The RMSE on SMOD shows that some errors still remain in the SMOD simulation, likely a result of errors in the meteorological forcings. However, the results of our study are mostly based on relative differences between simulations and not on absolute value, thereby reducing the impact of uncertainties in the simulated accumulation and ablation, as mentioned in the section dedicated to the limitations of the study. The evaluation of SMOD is impacted by errors in the precipitation inputs, whereas snow melt rates are less sensitive. This is why an evaluation of the simulated melt rate at the 495 automatic stations was provided in the first version of the manuscript.

The Method section (lines 364-368) was thus updated as follows: *"The simulations are evaluated using 495 snow depth measurements stations located in the French Alps and Pyrenees, spanning an elevation range from 1200 m to 2700 m asl, over the period 1983–2018. Two types of evaluation are carried out. First, we compare simulated and observed SMOD (Supplementary Table S1). Two error metrics were used: the root mean square error (RMSE) and the mean bias (a positive bias indicates a later simulated SMOD compared to the observation). Results show that considering the explicit representation of BC and dust outperforms the other simulation set-up with an overall bias of 1.7 days. However, SMOD is also strongly affected by uncertainties in the meteorological forcings. Therefore, an additional evaluation is carried out focusing on daily melt rates. The evaluation is done by comparing the simulated and measured daily variations of snow depth in case of melt (Queno et al., 2017, Supplementary text). Considering the explicit representation of BC and dust generally leads to a lower bias compared to the other simulations (Supplementary Fig. S8)."*

Table S1 was added in the supplementary material and reads :

Lines 73-74 in the main text were also modified as follows : *To quantify the impact of BC and dust on snow cover evolution we used the snow melt-out date (SMOD), defined as the last date of the annual longest period with at least 30 cm of snow. This indicator is relevant for water resources and ecosystem studies (Hock et al., 2019). The snow cover simulations were evaluated against satellite observations of the snow cover area and in situ observations of snow depth at 495 stations (See Methods). Simulations accounting for BC and dust systematically lead to better scores compared to pure snow simulations, with an overall SMOD bias of -1.7 days at the measurement stations.*

Simulations	Scores	All elevations
Baseline	RMSE (days)	26.3
	Bias (days)	3.8
	# Values	1028
BC + dust	RMSE (days)	25.7
	Bias (days)	-1.7
	# Values	1074
Pure	RMSE (days)	31.1
	Bias (days)	13.3
	# Values	928

Table 1: **Table S1: SMOD evaluation at the snow-depth measurement stations.** Scores between simulated SMOD and observed SMOD from 495 snow-depth measurement stations located in the French Alps and Pyrenees. Numbers in bold indicate the best score.

2. The methodology of segregation of LAP and warming effect on SMOD is not clear. More information should be given in the ANOVA analysis as understanding the analysis is hard in the present form. Actually, what is the response of SMOD to warming is not clear. The title says that LAPs alter this response. For instance, the sensitivity of SMOD to LAP can also changing with warming, how is it accounted?

We provide here a twofold response. The first part is on the ANOVA analysis and the segregation of LAP and meteorological conditions effects discussed in section **BC and dust effect on snow cover inter-annual variability** of the main text. The second part is on the sensitivity of SMOD to LAP and warming (section **Past trends (1979–2018)**).

First, we agree that the ANOVA analysis and the segregation between the effect of LAP and of meteorological conditions was not sufficiently detailed in the supplementary material and in the main text. The segregation between the effect of LAPs and the effect of meteorological conditions is done thanks to the ANOVA analysis but only for the variance of the SMOD (not for the SMOD absolute values). To separate the effects of LAP and of meteorological conditions, we used the methodology from Geoffroy et al., 2012. The SMOD, S , can be decomposed as a sum of one-variable functions and an interaction term I :

$$S = p_0 + p_1(\text{meteo}) + p_2(\text{LAP}) + I(\text{meteo}, \text{LAP}) \quad (6)$$

where p_0 is constant, p_1 depends on the meteorological conditions only, p_2 depends on LAPs deposition fluxes only and I is an interaction term, including first and higher order interactions between the meteorological conditions and LAPs deposition. We called I sequencing in our study. We extended the supplementary text on the ANOVA analysis and we fully revised lines 111-119 in the main text. The proposed changes are reported below, please see also response to referee 1 comment 9 for more details.

"A. ANOVA equations

Following Eq. 6 in Geoffroy et al., 2012, the SMOD, noted S , can be decomposed as a sum of one-variable functions and an interaction term I , referred to as a "sequencing" term:

$$S = p_0 + p_1(\text{meteo}) + p_2(\text{LAP}) + I(\text{meteo}, \text{LAP}) \quad (7)$$

where p_0 is constant, p_1 depends on the meteorological conditions only, p_2 depends on LAPs deposition fluxes only and I includes interactions between meteorological conditions and LAPs deposition, i.e., the fact that the response to given LAP deposition can depend on meteorological conditions. For example, dust might be directly buried after deposition if a sufficiently large snowfall event occurs immediately after dust deposition.

The contribution, \hat{c} , of each function p_1 , p_2 to the variance of the SMOD is computed following the equations (2) and (3), where N_1 denotes the number of values taken by the meteorological conditions (39 individual years) and $N_2 = 2$ (simulations with or without LAPs). i, j are the parameter values. The interaction term (\hat{c}_I) (or sequencing) is computed following the equation (4).

$$\hat{c}(p_1) = \frac{1}{\text{var}(S)} \frac{1}{N_1} \sum_{i=1, N_1} \left[\frac{1}{N_2} \sum_{j=1, N_2} (S_{i,j} - \bar{S}) \right]^2 \quad (8)$$

$$\hat{c}(p_2) = \frac{1}{\text{var}(S)} \frac{1}{N_2} \sum_{i=1, N_2} \left[\frac{1}{N_1} \sum_{j=1, N_1} (S_{i,j} - \bar{S}) \right]^2 \quad (9)$$

$$\hat{c}(I) = 1 - (\hat{c}(p_1) + \hat{c}(p_2)) \quad (10)$$

"

Lines 111-119 were modified as follows: *"For given deposition fluxes of BC and dust, the impact on the snow cover depends on the meteorological conditions (Tuzet et al., 2020). For instance, the same amount of BC and dust deposited at the snow surface has a stronger impact if a long time period without snowfall follows the deposition, than when a new snowfall buries the particles immediately after deposition. Using the ANOVA method, we can separate two types of contribution to the SMOD variance, the contribution of LAPs deposition observed for average weather conditions, and the contributions through interactions between meteorological conditions and LAP deposition. The later is named the sequencing part (see Supplementary text). This sequencing of meteorological conditions and BC and dust deposition contributes to 2.9 – 9.1% to the total variance, while the deposition only contributes to 12.7 – 25.5% of the total variance (Fig. 2). Hence, deposition of dust and BC markedly contributes to the variance of the SMOD and this is partly due to the sequencing between the meteorological conditions and the deposition."*

In the section **Past trends**, we analyse the response of SMOD to warming with and without LAPs. Figure 3 and Figure S3 show the SMOD trends with dust and BC, with BC only, dust only and without dust and BC. The SMOD trends calculated on the simulations without LAPs (blue markers) depict the response of SMOD to warming only if there was no LAPs deposition. The red markers show the SMOD trends accounting for LAPs deposition. The fact that the blue and the red markers differ demonstrates that LAP deposition changes the response of SMOD to warming. The sensitivity of SMOD is changing with warming and this is accounted for directly in the detailed snowpack model with the explicit hourly deposition of LAPs and hourly meteorological forcings. What we did not investigate, however, is the response of SMOD with LAPs but without warming. The entire section **Past trends** was reformulated to ease the reading. The changes are reported below (please see also modifications and response to comments 12 and 13 of reviewer 1).

"Past trends (1979–2018)

The simulated SMOD for snow with BC and dust shows a negative trend over the last decades (1979–2018) indicating earlier snowmelt (Fig. 3). In this section, we mostly discuss the results in the North Alps since this is the only area where most of the trends are significant. In the Northern Alps, the SMOD at 2100 m a.s.l. decreases by 5.7 days per decade (corresponding to a decrease of 2.9% of the snow season duration), which is consistent with Maitiu et al., 2021. This decrease is consistent with the atmospheric warming rate. This warming rate ranges between +0.25 and 0.3°C per decade according to the reanalysis data used in this study (Fig. 3).

The simulated SMOD trend for pure snow is more pronounced than for snow with BC and dust. At 2100 m a.s.l., the pure snow SMOD decrease is 7.8 days per decade for the North Alps. The deposition of dust and BC thus alters the response of snow to warming. First, the sensitivity of SMOD to meteorological conditions is not the same for pure snow and for snow containing LAPs (e.g. Fig. 8 in Dumont et al., 2020). Second, the dust and BC inputs were not constant over the last four decades. BC deposition exhibits a negative trend, with a decrease of 11% on average considering all the massifs. Simulations accounting for BC deposition yield a negative SMOD trend lower than the SMOD trend of the pure snow simulation (Fig. 3). This means that, at the beginning of the study period (1979–1988), the shortening attributed to BC deposition alone (i.e. 15 days on average for all the areas) is higher than for the recent period (2009–2018) (i.e. 10 days only in average). In this recent period, the deposition of BC is reduced compared to the 1980s. BC deposition accelerates melt less than in the 1980s. Thus, the reduction of BC deposition fluxes partly offsets the effect of rising temperature in the 2009-2018 period. Conversely, no significant trend in dust deposition is identified. Dust deposition consequently affects the SMOD trend less than BC deposition does (Fig. 3).

Hence, the deposition of BC and dust modifies the SMOD trend in response to climate warming. In the North Alps, the primary driver of the trend modification is the decrease in BC deposition since the 1980s. This decrease compensates part of the impact of warming on the trend of the snow cover duration at 2100 m a.s.l. and is also observed for all the Alps and Pyrenees to a lesser extent (Supplementary text and Fig. S3). However, the trends presented in this section originate from a single model and the limitations of this study are discussed below."

3. The estimates of LAP-induced changes in SMOD or CMD is 10%. These estimates are from offline simulations and do not include the aerosol-atmosphere and aerosol-snow feedbacks and couplings. Authors

have mentioned many aspects of meteorological uncertainty and also justified the estimations. But, a clear discussion on uncertainties due to offline mode of analysis (comparing their findings with LAP-snow coupled simulations) should be included. please see: Rahimi et al., 2020 ACP; Sarangi et al., 2019 ACP; Usha et al., 2021 ERL; Zhao et al., 2014; ACP etc

We agree, although this limitation was already mentioned in the discussion section of the manuscript (lines 237-239). The section was modified as follows to make it clearer :

"Finally, the impact of BC and dust deposition on snow is estimated from offline simulations and thus does not include neither the aerosol-atmosphere interactions nor the coupling between the surface and the atmosphere. Aerosols might cool the atmosphere by scattering the solar radiation before being deposited onto the surface, while snow cover reduction can lead to positive feedback enhancing the atmospheric warming by advection of turbulent heat (Liston, 1995, see Method Limitations). Further research is required to investigate the simultaneous responses of the snow cover to these processes especially the potentially compensating effect of the particles in the atmosphere. In this study, we thus estimated that the interval of confidence associated with the average SMOD changes attributed to BC and dust is ± 6 days."

A specific paragraph "Offline simulations" was also added in the limitations section in the Methods (line 511) with the suggested references and reads as follow : *"Offline simulations*

Simulations are performed in offline mode, i.e. the aerosol-atmosphere interactions and the coupling between the surface and the atmosphere are not considered. The enhanced melt estimated in this study could be partly compensated by the cooling effect of the aerosols in the atmosphere before their deposition. Indeed, for instance, Rahimi et al. (2020) quantified an enhanced melt due to BC of 3-12 kg m⁻² while the aerosol-radiation interactions dampen the surface of incoming solar energy to a level equivalent to the preservation of 1-5 kg m⁻² of snow. Zhao et al. (2014) showed an effect of similar magnitude of radiative warming in the snowpack and the magnitude of surface radiative cooling due to BC and dust in the atmosphere (10 W m²). On the opposite, not accounting for the coupling between the surface and the atmosphere coupling can result in an underestimation of the aerosol effects on snow (Sarangi et al., 2019; Usha et al., 2021). Further research is warranted to investigate the above-mentioned feedbacks at a regional scale in mountainous areas."

Reviewers' Comments:

Reviewer #1:

Remarks to the Author:

The authors have done a substantial revision. Most of my concerns have been resolved. Especially in the methodology, the authors cited recent published studies to discuss the uncertainties brought by the meteorological forcing such as the shortwave radiation, temperature and precipitation. Although the magnitude of SMOD change caused by these meteorological uncertainties was not presented, the authors provided further confidence in the results by the validation at the 495 stations. Generally I endorse the authors for carefully revising this work. Several minor points need to be addressed.

1. The authors need to clear the confidence interval of the SMOD change which is determined by the difference among stations or other factors.
2. The BC and dust contribution to the SMOD and snowmelt runoff are larger in the Alps than in the Pyrenees. Is the driving role of meteorological condition in the Pyrenees more obviously, or just for the less BC deposition.
3. In Figure 1, the box plot of baseline (considering implicitly the LAPs) seems blank but was described as green. Also, the caption of Figure S1 (Annual BC and dust deposition) doesn't match the figure (SMOD reduction). Please revise them and check details carefully.

Reviewer #2:

Remarks to the Author:

The authors have replied to many of my comments. However, as the novelty of the paper is about the quantification and not the process-level understanding, I am still concerned with the large RMSE in the SMOD evaluation included.

Given the low biases, RMSE represents the variance in the model prediction. The errors are in the range of 25-30 days, while the paper's main result is that SMOD shifts by ~ 15 days due to LAPs. The quantification is well within the RMSE range and thus within the variance noise.

Response to reviews

Black carbon and dust alter the response of mountain snow cover under climate change

Marion Reveillet, Marie Dumont, Simon Gascoin, Matthieu Lafaysse, Pierre Nabat, Aurelien Ribes, Rafife Nheili, Francois Tuzet, Martin Menegoz, Samuel Morin, Ghislain Picard, and Paul Ginoux

May 9, 2022

Introduction

We would like to thank again the reviewers for the careful evaluation of our work. The manuscript have been revised according to their comments. We clarified the confidence interval of the SMOD change origin. We extended the discussion on the differences between the Alps and the Pyrenees. We also modified Figure 1 and Figures S1, S8, S11 and S12. The SMOD evaluation at the measurement stations and error analysis have been refined demonstrating the statistical relevance of the results. A point-by-point response to each comment is provided below with the proposed changes in the manuscript.

In the present document, comments from the referees are reported in the blue boxes before the authors response. In the response we use *italic* fonts to quote text from the manuscript and **blue** to highlight changes. A track-changes version of the revised manuscript is also provided.

Reviewer #1 (Remarks to the Author)

The authors have done a substantial revision. Most of my concerns have been resolved. Especially in the methodology, the authors cited recent published studies to discuss the uncertainties brought by the meteorological forcing such as the shortwave radiation, temperature and precipitation. Although the magnitude of SMOD change caused by these meteorological uncertainties was not presented, the authors provided further confidence in the results by the validation at the 495 stations. Generally I endorse the authors for carefully revising this work. Several minor points need to be addressed.

The authors thank the reviewer for the time spent on the manuscript and the helpful comments. A detailed point-by-point response is provided below.

1. The authors need to clear the confidence interval of the SMOD change which is determined by the difference among stations or other factors.

We agree that the way the confidence interval was introduced could be a bit confusing. It was however mentioned at the end of the discussion (i.e lines 257-268) that it was determined by the uncertainty caused by the choice of the model parameters (dust refractive index, scavenging coefficient and snow model physics). A detailed evaluation of the simulation SMOD compared to the stations is now provided in the revised paper (section Methods and Tab. S1) in response to comment 1 of reviewer 2. Please see response to comment 1 of reviewer 2 for the proposed changes.

To clarify this point, we propose the following changes in manuscript :

line 267 : *"In this study, we thus estimated that the confidence interval associated with the average SMOD changes attributed to BC and dust is ± 6 days. This confidence interval represents the uncertainty (standard deviation) due to model parameters (refractive index, scavenging coefficient and model physics) as detailed above and in the Methods section."*

2. The BC and dust contribution to the SMOD and snowmelt runoff are larger in the Alps than in the Pyrenees. Is the driving role of meteorological condition in the Pyrenees more obviously, or just for the less BC deposition.

The table below sums up the ANOVA analysis for several elevations and for the 4 regions. It shows that the contribution of the meteorological conditions generally decreases with elevation for all the regions. This is in line with the increasing effect of light absorbing impurities on SMOD with elevation (e.g. shown on Figure 1). In addition, BC deposition in the Pyrenees is lower than in the Alps (Fig. S2) and snow melts overall earlier in the Pyrenees than in the Alps for a given elevation (Fig. S1). Finally, in the pure snow simulations, the SMOD standard deviation over the whole period is slightly higher in the Pyrenees (10%) than in the Alps (9%) (Fig. S1).

Elevation	Location	Parameter	Contribution
1800	North Alp	Meteo / BC + Dust	72.0 / 21.5%
		Meteo / BC only	78.2 / 16.1%
	South Alps	Meteo / BC + Dust	74.9 / 15.0%
		Meteo / BC only	83.0 / 9.0%
	North Pyrenees	Meteo / BC + Dust	89.5 / 5.3%
		Meteo / BC only	94.7 / 2.3%
	South Pyrenees	Meteo / BC + Dust	89.6 / 4.6%
		Meteo / BC only	95.3 / 2.0%
2100	North Alp	Meteo / BC + Dust	71.5 / 25.5%
		Meteo / BC only	78.1 / 19.3%
	South Alps	Meteo / BC + Dust	69.3 / 22.0%
		Meteo / BC only	80.2 / 12.4%
	North Pyrenees	Meteo / BC + Dust	79.8 / 13.9%
		Meteo / BC only	91.4 / 6.2%
	South Pyrenees	Meteo / BC + Dust	78.2 / 12.7%
		Meteo / BC only	89.1 / 5.9%
2400	North Alp	Meteo / BC + Dust	63.8 / 31.3%
		Meteo / BC only	71.0 / 23.9%
	South Alps	Meteo / BC + Dust	66.4 / 28.0%
		Meteo / BC only	77.8 / 17.4%
	North Pyrenees	Meteo / BC + Dust	68.7 / 24.1%
		Meteo / BC only	78.0 / 16.8%
	South Pyrenees	Meteo / BC + Dust	70.4 / 21.9%
		Meteo / BC only	82.4 / 12.0%

Table 1: Contribution of the meteorological conditions and BC and dust deposition to the variance of the SMOD. Contributions computed over the 1979–2018 period at 1800, 2100 and 2400 m a.s.l. (in complement of Figure 2) for the North and South Pyrenees and Alps. The sequencing term is not shown.

In summary, the simulations show that the BC and dust contributions to the SMOD and snowmelt runoff are larger in the Alps than in the Pyrenees as a result of (i) lower BC deposition (ii) earlier snowmelt and (iii) higher variability in meteorological conditions in the Pyrenees than in the Alps.

The discussion was inserted in the manuscript line 137 that now reads :

"The BC and dust contribution to the SMOD inter-annual variability is larger in the Alps than in the Pyrenees. This is due to the combined effects of three different factors: lower BC deposition (Fig. S2), earlier snowmelt and slightly higher inter-annual variability of the meteorological conditions in the Pyrenees than in the Alps (Fig. S1). BC deposition contributes "

3. In Figure 1, the box plot of baseline (considering implicitly the LAPs) seems blank but was described as green. Also, the caption of Figure S1 (Annual BC and dust deposition) doesn't match the figure (SMOD reduction). Please revise them and check details carefully.

Thank you for pointing out this mistake, the caption of Fig. 1 has been corrected as follow:

"Shortening of the snow season related to BC and dust deposition This shortening is computed as the difference in snow melt-out date (SMOD) between the pure snow simulations and all the configurations: simulations

considering BC and dust (brown), only BC (grey), only dust (orange), considering implicitly the LAPs (i.e. an albedo decrease based on the snow age only (white, hatched)). SMOD differences are computed as a function of the elevation, considering the entire study period (i.e., 1978–2018), for the North and South Pyrenees (left) and the North and South Alps (right). The boxes show the quartiles of the distribution corresponding to the inter-annual and spatial variability. Minimum/maximum ranges (excluding outliers) are indicated by the whiskers. Only elevations with a mean SD >30 cm over the winter period (i.e. 1st of December to 30 of April) are represented."

To be consistent with Figure 1, Figures S8, S11 and S12 have been revised: the baseline is represented by white hatched boxes (instead of green boxes). The caption has been modified accordingly (see below). Finally, all the figures (including in the supplementary material) have been checked carefully.

Figure 1: Figure S8: Model evaluation using station measurements. Number of values, RMSE and bias computed over the 1983–2018 period, between the daily observed corresponding to melt and simulated ΔSD corresponding to melt for $S_{BC+Dust}$ (brown), S_{pure} (blue) and $S_{baseline}$ (white, hatched). Results are presented by elevation range for the French Alps (left) and the Pyrenees (right) separately.

Figure 2: Figure S11: Shortening of the season computed as the difference in snow melt-out date (SMOD) between the pure snow simulations and simulations considering BC and dust (red), dust only (orange), BC only (grey) and considering implicitly the LAPs (i.e. an albedo decrease based on the snow age only, baseline, (white, hatched)). SMOD differences are computed at 2100 m a.s.l., considering the entire study period (i.e., 1978-2018), for each individual SAFRAN massif in the French Alps - see Fig. S7. The boxes show the quartiles of the distribution corresponding to the inter-annual variability. Minimum/maximum ranges (excluding outliers) are indicated by the whiskers.

Figure 3: Figure S12: Shortening of the season computed as the difference in snow melt-out date (SMOD) between the pure snow simulations and simulations considering BC and dust (red), dust only (orange), BC only (grey) and considering implicitly the LAPs (i.e. an albedo decrease based on the snow age only, baseline, (white, hatched)). SMOD differences are computed at 2100 m a.s.l., considering the entire study period (i.e., 1978-2018), for each individual SAFRAN massif in the Pyrenees - see Fig. S7. The boxes show the quartiles of the distribution corresponding to the inter-annual variability. Minimum/maximum ranges (excluding outliers) are indicated by the whiskers.

Reviewer #2 (Remarks to the Author)

The authors have replied to many of my comments. However, as the novelty of the paper is about the quantification and not the process-level understanding, I am still concerned with the large RMSE in the SMOD evaluation included.

We understand the reviewer concern. To strengthen the conclusions, we analysed the SMOD errors at the measurement stations more deeply. This error analysis demonstrates that the Δ SMOD attributed to BC and dust calculated from the simulations is greater than the uncertainty of the model itself and is detailed below.

Given the low biases, RMSE represents the variance in the model prediction. The errors are in the range of 25-30 days, while the paper’s main result is that SMOD shifts by 15 days due to LAPs. The quantification is well within the RMSE range and thus within the variance noise.

A summary of the error analysis presented below is provided at the end of the response to this comment.

First of all, we identified a problem with the observation dataset where missing values were incorrectly filtered out, leading to extreme differences between the simulation and observation (typically higher than 50 days). These observations were removed from the calculation leading to an important decrease in RMSE and bias. Note that this only impacts the SMOD evaluation, and not the other evaluations presented in the paper. The new results are reported in blue below close to the old ones in black. To characterise in more detail the error distribution, the new Table 1 also provides the values of the mean absolute error, MAE and the interquartile range of the error, i.e. the distance between the first and the third quartiles.

Simulations	Scores	All elevations
Baseline	RMSE (days)	19.4 26.3
	MAE (days)	12.8
	IQ range (days)	16.0
	Bias (days)	3.6 3.8
	# Values	788 1028
BC + dust	RMSE (days)	19.2 25.7
	MAE (days)	13.3
	IQ range (days)	18.5
	Bias (days)	-0.68 -1.7
	# Values	803 1074
Pure	RMSE (days)	23.6 31.1
	MAE (days)	17.0
	IQ range (days)	21.5
	Bias (days)	12.4 13.3
	# Values	715 928

Table 2: **SMOD evaluation at the snow depth measurements stations.** Scores between simulated SMOD and observed SMOD from 495 stations located in the French Alps and Pyrenees. Numbers in bold indicate the best score. Four metrics are indicated, the root mean square error (RMSE), the mean absolute error (MAE), the interquartile range (IQ) and the bias (model-observation). The interquartile range is given as the distance between 25% and 75% quantiles of the error.

Although MAE and RMSE (respectively of 13 and 19 cm) seem to be highly influenced by some locally high errors, more than half of the SMOD errors actually lie within ± 10 days (half of the interquartile range of the error, Tab.), which is smaller than the estimated impact of BC and dust (17 days). By definition, the RMSE is strongly influenced by extreme values, typically outside the interquartile range. These large differences between observed and simulated SMOD are most of the time related to snow free day that occurs during the winter season in one of the time series. In the study, we define the SMOD as the day of free snow surface after the longest period with at least 30 cm of snow. When a snow free day occurs in the middle of the winter, this snow free day will then be taken as SMOD and this can lead to high values of SMOD errors as illustrated in figure 4). When calculating the RMSE only accounting SMOD values later than March 1st to avoid the extreme values explained above, the BC+dust simulation drops from 19.2 to 15.7 days from RMSE and for 13.3 to 11.4 days for MAE.

Figure 4: Simulated (orange) and observed (blue) snow depth evolution at one station over the 1988-1989 winter season

The RMSE of 19.2 days on SMOD for the BC+dust simulations is not surprising since the evaluation is done at point locations where local variability can be high (local topography effect, snow drift ...). This is also shown and discussed in Vernay et al., 2021, where Figures 12 and 13 describe the bias and root mean square deviation of total snow depth using the same snow depth measurement stations as in our study. The detailed local evaluations shown in Tuzet et al., 2017 (Figure 3), Dumont et al., 2020 (Figure 5) ; Tuzet et al., 2020 (Figure 5) show that when the meteorological forcing is observed, the local simulations of snow melt rate and melt out date is perfectly fitting with the observations and the SMOD error lies within $\pm 1-2$ days. This shows that most of the RMSE in Table S1 likely comes from the spatial scale discrepancy between the meteorological forcing and snow observations, and possibly other residual errors in the meteorological forcing. This RMSE of 19 days is the averaged error for one station, one year. On the opposite, the 17 days shortening effect mentioned as one of the main conclusions of the paper is a 40 year average, integrated over the full area of the Alps and the Pyrenees, most likely smoothing out the local and annual topographic effects.

In addition, the difference in SMOD between the BC+dust simulation and the pure snow simulation is significant for bias, RMSE and MAE. To demonstrate this, we performed bootstrap experiments on the difference between the simulated and the observed SMOD. We randomly draw 35 years with replacement over the 1983-2018 period, which includes most of the in-situ observations. We therefore obtain 35 random years, and as the draw is with replacement, years can be drawn several times, or not drawn. The random draw is performed 1000 times for both the pure and BC+dust simulations, in order to obtain 1000 series of 35 years. Then we computed the mean bias and the RMSE for each series with respect to the in situ data, to obtain 1000 values of bias and RMSE. Figure 5 shows the distribution of the 1000 biases for both pure snow simulation and BC+dust simulation, and, as there is no overlap between the two distributions, it demonstrates that the biases of these two simulations are significantly different. Regarding the RMSE, the two distributions do show some overlap (not shown), but the distribution of the difference in RMSEs (fig. 6) is significantly positive, which means that the difference in RMSEs between the pure and the BC+dust simulations is significant. The same results stand for the MAEs (not shown). By showing the significance of the difference of the bias, RMSE and MAE of the pure snow simulation and BC+Dust simulation, we can conclude that the reported averaged shortening of 17 days is greater than the uncertainties due to the model itself.

Figure 5: Distribution of bias between observed and simulated SMOD for pure simulation (blue) and BC+dust simulation (brown) at the weather station. The distribution corresponds to a bootstrap experiment on the difference between the simulated and the observed SMOD with 1000 random draws over the 1983-2018 period.

Figure 6: Distribution of the RMSE difference between a pure simulation and a simulation accounting for BC+dust at the weather station. To obtain this distribution, first the RMSE between the observed SMOD and the simulated SMOD for the pure and BC+dust simulations are calculated with a bootstrap experiment with 1000 random draws over the 1983-2018 period. Then the difference between the RMSE obtained for pure simulations and BC+dust simulation is calculated. This figure shows the 95% confidence interval of the distribution.

Summary of the error analysis

- The RMSE of the model at all the observation stations is 19 days likely mostly due to the local variability of the meteorological variables at the scale of the observation stations. The reported 17 days shortening effect of BC and dust is a 40 year average integrated over the full area of the Alps and the Pyrenees, most likely smoothing out the local and annual effects.
- Model evaluation using accurate meteorological forcing at the station scale (Tuzet et al., 2017, 2020; Dumont et al., 2020) shows that the SMOD simulations are accurate within $\pm 1-2$ days.
- The RMSE of 19 days is strongly influenced by extreme values, e.g. most of the time due to one snow free day during the winter and decreases to 15.7 days when calculated starting on March 1st only. The

MAE value (13.3 days) is lower than the 17 days shortening attributed to the BC and dust deposition.

- The difference in bias, RMSE and MAE between the pure snow and the BC+dust simulations are statistically significant implying that the Δ SMOD attributed to BC and dust is greater than the uncertainties of the model itself.

The reviewer is right that the effect of the LAPs on the SMOD has approximately the same magnitude as the residuals variance. However it does not necessarily mean that the effect is not significant compared to the model error. Indeed, the effect of the LAPs on the SMOD is not random since it systematically tends to shift simulations towards earlier melt (with a variability depending on the site and the year). Therefore it causes a significant shift of the residual distribution mean as illustrated by Figure 5 from the bootstrap experiment.

Proposed changes in the manuscript

To account for this analysis, the manuscript was modified as follows :

Line 84 now reads : *"Simulation accounting for BC and dust systematically lead to better scores compared to pure snow simulations, with an overall SMOD bias of -0.68 days, an RMSE of 19.2 days and an MAE of 13.3 days at the measurement stations (see Methods section)."*

The section devoted the comparison with snow depth measurements line 391 in the Methods section was rewritten as follows :

"The simulations are evaluated using 495 snow depth measurements stations located in the French Alps and Pyrenees, spanning an elevation range from 1200 m to 2700 m asl, over the period 1983–2018. Two types of evaluation are carried out. First, we compare simulated and observed SMOD (Supplementary Table S1). Three error metrics were used: the root mean square error (RMSE), the mean absolute error (MAE) and the mean bias (a positive bias indicates a later simulated SMOD compared to the observation). Results show that considering the explicit representation of BC and dust outperforms the pure snow simulation set-up with an overall bias of -0.68 days, a RMSE of 19.2 days and a MAE of 13.3 days. The RMSE is influenced by extreme values, frequently due to one snow free day in early winter in the observed time series. The RMSE calculated only for SMOD later than March 1st drops to 15.7 days (MAE 11.4 days). In addition, model evaluation using observed meteorological forcing at the station scale (Tuzet et al., 2017, 2020; Dumont et al., 2020) shows that the SMOD simulations is accurate within \pm 1-2 days. Most of the RMSE in Table S1 likely comes from the spatial scale discrepancy between the meteorological forcing and snow observations, and possibly other residual errors in the meteorological forcing. To investigate the statistical significance of the differences in scores between the pure and the BC+dust simulations, we use a bootstrap experiment. Namely, we randomly draw 35 years with replacement over the 1983-2018 period, which includes most of the in-situ observations. We therefore obtain 35 random years, and as the draw is with replacement, years can be drawn several time, or not drawn. The random draw is performed 1000 times for both the pure and BC+dust simulations, in order to obtain 1000 series of 35 years, providing 1000 values of bias, RMSE and MAE. The bias, RMSE and MAE distributions show that the difference in bias, RMSE and MAE between the pure snow and the BC+dust simulations are statistically significant. This implies that the Δ SMOD attributed to BC and dust is greater than the uncertainties of the model itself. Since SMOD is strongly affected by uncertainties in the meteorological forcing, an additional evaluation is carried out focusing on daily melt rates. The evaluation is done by comparing the simulated and measured daily variations of snow depth in case of melt (Queno et al., 2017, Supplementary text). Considering the explicit representation of BC and dust generally leads to a lower bias compared to the other simulations (Supplementary Fig. S8)."

Table S1 was changed to :

Simulations	Scores	All elevations
Baseline	RMSE (days)	19.4
	MAE (days)	12.8
	IQ range (days)	16.0
	Bias (days)	3.6
	# Values	788
BC + dust	RMSE (days)	19.2
	MAE (days)	13.3
	IQ range (days)	18.5
	Bias (days)	-0.68
	# Values	803
Pure	RMSE (days)	23.6
	MAE (days)	17.0
	IQ range (days)	21.5
	Bias (days)	12.4
	# Values	715

Table 3: Table S1: **SMOD evaluation at the snow-depth measurements stations.** Scores between simulated SMOD and observed SMOD from 495 stations located in the French Alps and Pyrenees. Numbers in bold indicate the best score. Four metrics are indicated, the root mean square error (RMSE), the mean absolute error (MAE), the interquartile range (IQ) and the bias (model-observation). The interquartile range is given as the distance between 25% and 75% quantiles of the error.

Reviewers' Comments:

Reviewer #1:

Remarks to the Author:

Most of my concerns have been addressed to my satisfaction. Here only one minor one: The title of Figure S1 (Annual BC and dust deposition) seems not have been modified, it doesn't match the figure (SMOD reduction).

Reviewer #2:

Remarks to the Author:

The authors have attempted various possible ways to justify the concern related to the bias/RSME in simulated SMOD being almost similar to the BC-induced SMOD change and have included a paragraph in the revised manuscript accordingly. I appreciate their effort for the same.

In the process, they have illustrated that the biases are mostly due to the huge meteorological uncertainties involved in simulating SMOD (if observed meteorology is used, the simulated values have very less bias). This actually may lead to another question about the fidelity on the ANOVA based quantification of meteorology and LAP contribution at mastiff scale. Please include the explanation for the same also.

I suggest that the uncertainty in the quantification of BC and dust impacts on trends and SMOD to be expressed in abstract too.

Response to reviews and information for the editor

Black carbon and dust alter the response of mountain snow cover under climate change

Marion Reveillet, Marie Dumont, Simon Gascoin, Matthieu Lafaysse, Pierre Nabat, Aurelien Ribes, Rafife Nheili, Francois Tuzet, Martin Menegoz, Samuel Morin, Ghislain Picard, and Paul Ginoux

June 27, 2022

We would like to thank again the reviewers for the careful reading of our revised version. The manuscript has been slightly modified according to the few comments. We provided more details about the forcing uncertainty impact and better justified the choice to work at a regional scale. We also modified the S1 caption. A point-by-point response to the comments is provided below with the proposed changes in the manuscript. Comments from the referees are reported in the blue boxes above our response. We use *italic* fonts to quote text from the manuscript and **blue** to highlight changes. A track-changes version of the revised manuscript is also provided.

Otherwise, the author checklist has been completed and is also provided. Data used in the study have been published in an open-source platform: <https://doi.org/10.5281/zenodo.6760050>

Finally, as requested you will find below a brief summary of the main findings of the paper:

- Black carbon and dust deposition advanced the end of the snow season by 17 days on average over the last 40 years in the French Alps and the Pyrenees.
- The snow cover decline due to the current climate warming was partly offset by the decrease in black carbon deposition observed since the 1980s.

Please use the following Twitter handles: @ReveilletM @mpneige @sgascoin @OSUG_fr @meteofrance @CNRS @CesbioLab @IGE

Responses to Reviewer #1 (Remarks to the Author)

Most of my concerns have been addressed to my satisfaction. Here only one minor one: The title of Figure S1 (Annual BC and dust deposition) seems not have been modified, it doesn't match the figure (SMOD reduction).

We thank the reviewer for the time spent on the manuscript, the careful reading of the revised version, and for pointed out this mistake. The title and the caption of Figure S1 have been corrected in the revised manuscript: ***Simulated snow melt-out date. Temporal evolution of the simulated snow melt out date (solid lines) over the 1979–2018 period, at 2100 m a.s.l., with the standard deviation (shaded areas, representing the spatial variability). Pure simulations are represented in blue and simulations considering the effect of BC and dust are in red.***

Responses to Reviewer #2 (Remarks to the Author)

The authors have attempted various possible ways to justify the concern related to the bias/RSME in simulated SMOD being almost similar to the BC-induced SMOD change and have included a paragraph in the revised manuscript accordingly. I appreciate their effort for the same.

We thank the reviewer for the time spent on the manuscript and the careful evaluation of our work. The reviewer comments helped us a lot improve the manuscript.

In the process, they have illustrated that the biases are mostly due to the huge meteorological uncertainties involved in simulating SMOD (if observed meteorology is used, the simulated values have very less bias). This actually may lead to another question about the fidelity on the ANOVA based quantification of meteorology and LAP contribution at massif scale. Please include the explanation for the same also.

In the manuscript, the reported model errors on snow depth were computed by comparing the simulated and the observed SMOD at the point scale (snow depth stations). As discussed in the last round of reviews, the RMSE and MAE are strongly influenced by local variability such as the local topography effects (see method section) and may not be representative of the model error at the massif scale as shown by the comparison with remote sensing data. This is why the ANOVA study was performed at a regional scale (North and South Alps and Pyrenees). The information has been added in the method section 'Strategy of BC and dust contribution quantification'. You can read in the revised manuscript:

The BC and dust contribution is quantified at a regional scale, considering four regions: North and South Alps and Pyrenees. The choice to work at a regional scale limits the uncertainties of SAFRAN reanalysis compared to the uncertainties estimated at the local scale of a station that are mainly impacted by local effects.

At the massif scale, the uncertainty was evaluated by comparing the observed snow cover area to satellite images (see method section and Figure S7 of the Supp Mat). For the simulation considering BC and dust, the RMSE is ranging between 9% and 18.3% depending on the massif, with a mean of 12.3% and 13.8% for the North and South Alps and 13.5% and 15.9% for the North and the South Pyrenees respectively. These specific RMSE values for the four studied areas are now reported in the supplementary material:

Supplementary Fig. S7 shows the scores for each massif when comparing observed SCA (from MODIS) and simulated SCA ($S_{BC+Dust}$, $S_{baseline}$ and S_{pure}), computed over the period 2000–2016, considering only the winter period (i.e., November to June). For bias and J, the mean is reported. Despite low and not significant differences for R^2 and J between the four simulations, for both the Alps and the Pyrenees, the explicit representation of BC and dust ($S_{BC+Dust}$) leads to a significant decrease in bias as well as a lower RMSE (Supplementary Fig. S7). Indeed, for the simulation considering BC and dust, the RMSE is ranging between 9% and 18.3% depending on the massif, with a mean of 12.3% and 13.8% for the North and South Alps and 13.5% and 15.9% for the North and the South Pyrenees respectively

Finally, we agree that SAFRAN reanalysis is prone to uncertainties, as discussed in the manuscript (section "Method limitation - Forcing Uncertainties").

We are therefore confident that, as (i) the forcing is capturing the temporal variability and (ii) the uncertainties are related to local effect and the evaluation at larger-scale shows lower uncertainties, the bias mentioned by the reviewer does not affect the conclusion of the ANOVA study.

I suggest that the uncertainty in the quantification of BC and dust impacts on trends and SMOD to be expressed in abstract too.

We fully agree that providing uncertainties on our results is very important. We indicated in the abstract the uncertainty on the advance of the snowmelt date (± 6 days). However, in this journal the abstract is limited to 150 words and we do not see how to convey an accurate message on the uncertainties on the trends in so few words. In addition, we did not write a trend value in the abstract but only a qualitative assessment. There are two specific sections in the manuscript (i.e. 'limitation' section in the main body and 'methods limitations') where we carefully discussed the uncertainties to make sure that the reader is well informed about the limitations of our study.